# Control of spinal motor neuron terminal differentiation through sustained *Hoxc8* gene activity

Catarina Catela[1,2], Yihan Chen[1,2], Yifei Weng[1,2], Kailong Wen[1,2], Paschalis Kratsios[1,2]*

[1]Department of Neurobiology, University of Chicago, Chicago, United States; [2]University of Chicago Neuroscience Institute, Chicago, United States

**ABSTRACT** Spinal motor neurons (MNs) constitute cellular substrates for several movement disorders. Although their early development has received much attention, how spinal MNs become and remain terminally differentiated is poorly understood. Here, we determined the transcriptome of mouse MNs located at the brachial domain of the spinal cord at embryonic and postnatal stages. We identified novel transcription factors (TFs) and terminal differentiation genes (e.g. ion channels, neurotransmitter receptors, adhesion molecules) with continuous expression in MNs. Interestingly, genes encoding homeodomain TFs (e.g. HOX, LIM), previously implicated in early MN development, continue to be expressed postnatally, suggesting later functions. To test this idea, we inactivated *Hoxc8* at successive stages of mouse MN development and observed motor deficits. Our in vivo findings suggest that *Hoxc8* is not only required to establish, but also maintain expression of several MN terminal differentiation markers. Data from in vitro generated MNs indicate *Hoxc8* acts directly and is sufficient to induce expression of terminal differentiation genes. Our findings dovetail recent observations in *Caenorhabditis elegans* MNs, pointing toward an evolutionarily conserved role for Hox in neuronal terminal differentiation.

*For correspondence: pkratsios@uchicago.edu

Competing interest: The authors declare that no competing interests exist.

## Editor's evaluation

This manuscript will be of interest to developmental geneticists interested in neuroscience, and how spinal motor neurons maintain their unique identities in adulthood after fate decisions are made in the embryo. The work here demonstrates that a Hox transcription factor acts as a terminal selector to control motor neuron identity, thus mirroring recent studies in *C. elegans*, and thus pointing towards this type of gene regulation as important in building diverse nervous systems.

## Introduction

Motor neurons (MNs) represent the main output of our central nervous system. They control both voluntary and involuntary movement and are cellular substrates for several degenerative disorders (*Arora and Khan, 2021*). Due to their stereotypic cell body position, easily identifiable axons and highly precise synaptic connections with well-defined muscles, MNs are exceptionally well characterized in all major model systems. Extensive research over the past decades in worms, flies, and mice has focused on the early steps of MN development, thereby advancing our understanding of the molecular mechanisms controlling specification of progenitor cells and young postmitotic MNs, as well as motor circuit assembly (*Osseward and Pfaff, 2019*, *Philippidou and Dasen, 2013*; *Sagner and Briscoe, 2019*; *Thor and Thomas, 2002*). In the vertebrate spinal cord, progenitor cell specification critically depends on morphogenetic signals, whereas initial fate determination of postmitotic MNs

relies on combinatorial activity of different classes of transcription factors (TFs) (*di Sanguinetto et al., 2008*, *Jessell, 2000*; *Lee and Pfaff, 2001*; *Stifani, 2014*). The focus in early development, however, has left poorly explored the molecular mechanisms that control the final steps of MN differentiation. Once MNs are born and specified, how do they acquire their terminal differentiation features, such as neurotransmitter (NT) phenotype, electrical, and signaling properties? And perhaps most important, what are the mechanisms that ensure maintenance of such features throughout life?

The terminal differentiation features of every neuron type are determined by the expression of specific sets of proteins, such as NT biosynthesis components, NT receptors, ion channels, neuropeptides, signaling molecules, transmembrane receptors, and adhesion molecules (*Hobert, 2008*). The genes coding for these proteins ('terminal differentiation genes') are continuously expressed from development through adulthood, thereby determining the functional and phenotypic properties of individual neuron types (*Hobert, 2008*; *Hobert, 2011*). Therefore, the challenge of understanding how MNs acquire and maintain their functional features lies in understanding how the expression of MN terminal differentiation genes is regulated over time. Importantly, defects in expression of such genes constitute one of the earliest molecular signs of MN disease (*Nutini et al., 2011*; *Shibuya et al., 2011*). However, the regulatory mechanisms that induce and maintain expression of terminal differentiation genes in spinal MNs are poorly defined. In part, this is due to: (a) a scarcity of temporally controlled gene inactivation studies that remove the activity of MN-expressed regulatory factors (e.g. TF, chromatin factor) at different life stages, and (b) a paucity of terminal differentiation markers for spinal MNs. Although recent RNA-Sequencing (RNA-Seq) studies have begun to address the latter (*Blum et al., 2021*; *Delile et al., 2019*; *Alkaslasi et al., 2021*), most genetic and molecular profiling studies on spinal MNs are not conducted in a longitudinal fashion, i.e., at embryonic and postnatal stages. Hence, how these cells become and remain terminally differentiated remains unclear.

To elucidate the molecular mechanisms that enable spinal MNs to acquire and maintain their terminal differentiation features, we took advantage of the orderly anatomical relationship between MN cell body location and muscle innervation, referred to as 'topography' (*Dasen and Jessell, 2009*). In the spinal cord, this topographic relationship is mostly evident along the rostrocaudal axis, where MN populations located in different spinal cord domains (e.g. brachial, thoracic, lumbar, sacral) innervate different muscles. In this study, we focused on the brachial domain, where postmitotic MNs are organized into two columns: (a) the lateral motor column (LMC) contains limb-innervating MNs necessary for reaching, grasping, and locomotion, and (b) the medial motor column (MMC) contains axial muscle-innervating MNs required for postural control (*Philippidou and Dasen, 2013*). Through a longitudinal RNA-Seq approach, we identified multiple terminal differentiation markers and novel TFs with continuous expression in embryonic and postnatal brachial MNs. Interestingly, we also found that several homeodomain TFs (HOX, LIM) that were previously implicated in the early steps of brachial MN development (e.g. initial specification, circuit assembly) (*Philippidou and Dasen, 2013*; *Stifani, 2014*) continue to be expressed in postnatal MNs. We therefore hypothesized that some of these TFs play additional roles in later steps of brachial MN development.

To test this hypothesis, we focused on Hox proteins because recent findings in the ventral nerve cord (equivalent to mouse spinal cord) of the nematode *Caenorhabditis elegans* identified Hox proteins as critical regulators of cholinergic MN terminal differentiation (*Feng et al., 2020*; *Kratsios et al., 2017*). Among the seven Hox genes retrieved from our RNA-Seq, *Hoxc8* is highly expressed both in embryonic and postnatal brachial MNs. A previous study showed that *Hoxc8* acts early to establish brachial MN connectivity (*Catela et al., 2016*). Here, we report a new role for *Hoxc8* in later stages of mouse MN development. By inactivating *Hoxc8* at successive developmental stages, we found that it is necessary for the establishment and maintenance of select terminal differentiation features of brachial MNs. Mechanistically, Hoxc8 acts directly to induce expression of terminal differentiation genes. Similar to our observations in brachial MNs, we identified additional Hox genes with continuous expression in thoracic and lumbar MNs, suggesting maintained Hox expression in MNs is a broadly applicable theme to other rostrocaudal domains of the spinal cord. Because Hox genes are also expressed in the mouse and human brain during embryonic and postnatal stages (*Lizen et al., 2017*; *Takahashi et al., 2004*; *Hutlet et al., 2016*; *Krumlauf, 2016*), similar Hox-based mechanisms to the one described here may be widely used in the nervous system for the control of neuronal terminal differentiation.

## Results

### Molecular profiling of mouse brachial MNs at embryonic and postnatal stages

We first sought to define the molecular profile of brachial MNs at embryonic and postnatal stages with the goal of identifying putative terminal differentiation markers for these cells. This longitudinal approach focused on postmitotic MNs at embryonic day 12 (e12) and postnatal day 8 (p8). We chose e12 because: (i) spinal e12 MNs begin to acquire their terminal differentiation features, such as NT phenotype (*Martinez et al., 2012*), and (ii) MN axons at e12 have exited the spinal cord (*Catela et al., 2016*). We chose p8 because: (i) these are several days after neuromuscular synapse formation (*Gautam et al., 1996*), and (ii) pups at p8 become more active, indicating spinal MN functionality. To genetically label e12 MNs, we used the *Mnx1-GFP (green fluorescent protein)* reporter mouse (*Wichterle et al., 2002*) as it primarily labels embryonic MNs at e12 (*Amin et al., 2015*; *Hanley et al., 2016*; *Sawai et al., 2022*; *Wichterle et al., 2002*; *Figure 1A*). Due to low expression of *Mnx1-GFP* at postnatal stages, we turned to an alternative labeling strategy and crossed *Chat^IRESCre* mice (*Rossi et al., 2011*) with the *Ai9* Cre-responder line (Rosa26-CAG^promoter^-loxP-STOP-loxP-tdTomato) (*Madisen et al., 2010*). At p8, we observed fluorescent labeling of spinal MNs with tdTomato (*Figure 1A*, *Figure 1—figure supplement 1*). Taking advantage of the topographic MN organization along the rostrocaudal axis, we followed a region-specific approach focused on the brachial region (segments C4-T1) that contains MNs of the MMC and LMC. Upon precise microdissection of this region (see Materials and methods), we used fluorescence-activated cell sorting (FACS) to isolate GFP-labeled brachial MNs from e12 *Mnx1-GFP* mice and tdTomato-labeled brachial MNs from p8 *Chat^IRESCre^::Ai9* mice (*Figure 1A*). Through RNA-Seq, we obtained and compared the molecular profile of these cells (see Materials and methods). We identified differentially expressed transcripts (>fourfold, p<0.05) in the e12 (3715 transcripts) and p8 (3209 transcripts) dataset (*Figure 1B*, *Supplementary file 1*), suggesting gene expression profiles of embryonic and postnatal brachial MNs differ. Two factors that could contribute to these transcriptional differences between the e12 and p8 datasets are: (1) different levels of gene expression (see next section), and (2) a small fraction of the FACS-sorted cells are not MNs. Indeed, *Mnx1* and *Chat*, in addition to MNs, are also expressed in small, nonoverlapping neuronal populations in the spinal cord (*Wilson et al., 2005*; *Zagoraiou et al., 2009*; *Wichterle et al., 2002*; *Figure 1—figure supplement 1*).

Subsequent gene ontology (GO) analysis on proteins from embryonically enriched (e12) transcripts revealed an overrepresentation of molecules associated with neuronal development, such as regionalization, dendrite formation, and axon guidance (*Figure 1C*, *Supplementary file 2*). Notably, the most enriched class of proteins in the e12 dataset is TFs, many of which are known to control MN development (*Figure 1D*, see next section). On the other hand, GO analysis on proteins from postnatally enriched (p8) transcripts uncovered an overrepresentation of molecules associated with cell metabolism, such as ATP synthesis, oxidative phosphorylation, and energy-coupled proton transport (*Figure 1C–D*, *Supplementary file 2*), perhaps indicative of the higher metabolic demands of p8 MNs compared to their embryonic (e12) counterparts.

To identify terminal differentiation markers with continuous expression in brachial MNs, we leveraged our e12 RNA-Seq dataset (*Figure 1D*, *Supplementary file 1*). We arbitrarily selected eight genes coding for NT receptors, ion channels, and signaling molecules (*Slc10a4, Nrg1, Nyap2, Sncg, Ngfr, Glra2, Cldn1, Cacna1g*) and evaluated their expression at different life stages. Through RNA ISH, we found six genes (*Slc10a4, Nrg1, Nyap2, Sncg, Ngfr, Glra2*) with continuous expression in putative brachial MNs at embryonic (e12) and early postnatal (p8) stages (*Table 1*, *Supplementary file 1*). Available RNA ISH data from the Allen Brain Atlas also confirmed their expression at p56 (*Table 1*). The ventrolateral location of the cells expressing these six genes in the spinal cord strongly suggests they constitute terminal differentiation markers for brachial MNs.

### Developmental transcription factors continue to be expressed in spinal MNs at postnatal stages

Two simple, but not mutually exclusive mechanisms can be envisioned for the continuous expression of terminal differentiation genes (e.g. *Slc10a4, Nrg1, Nyap2, Sncg, Ngfr, Glra2*) in brachial MNs. Their embryonic initiation and maintenance could be controlled by separate mechanisms involving distinct

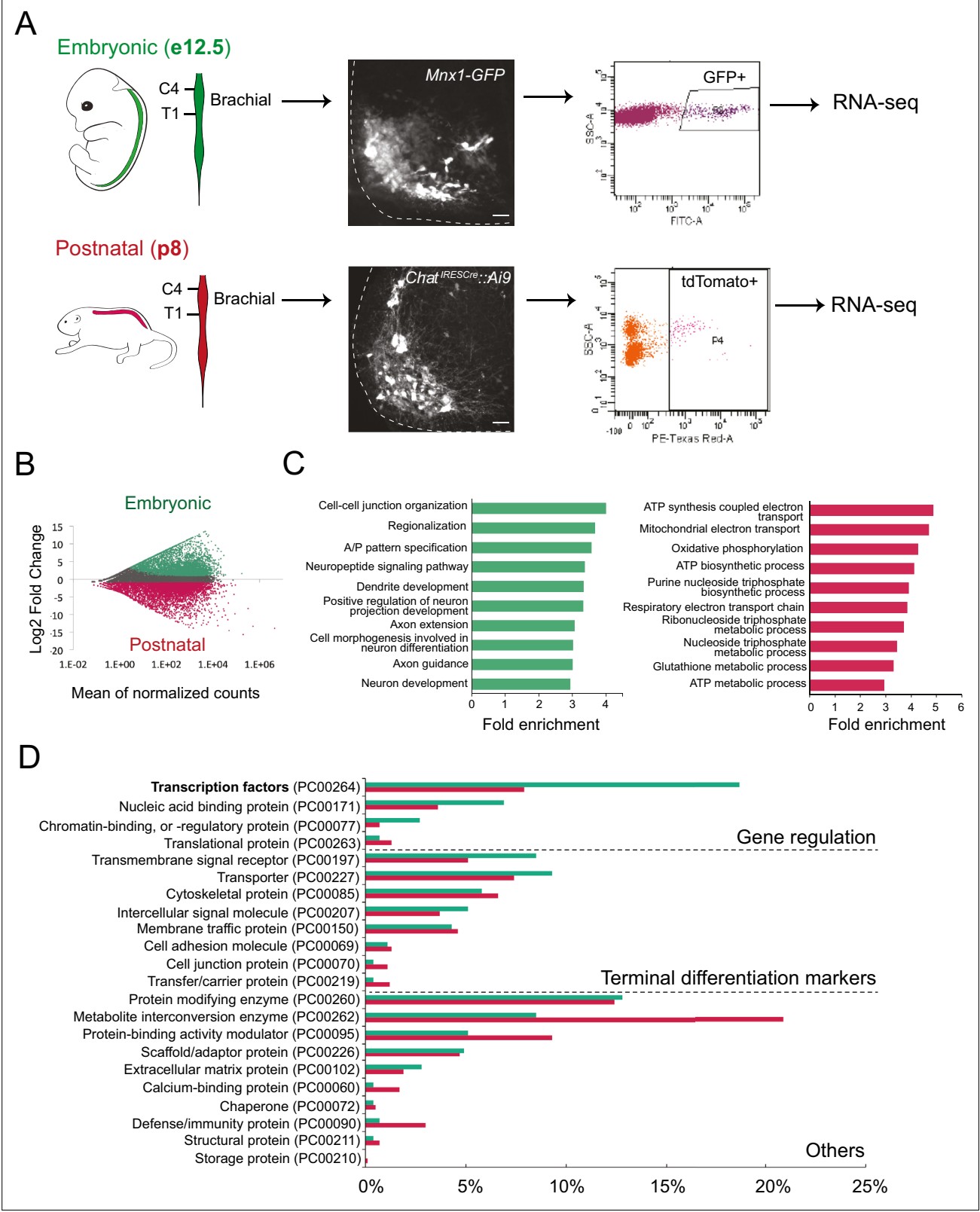

**Figure 1.** Molecular profiling of mouse brachial motor neurons (MNs) at embryonic and postnatal stages. (**A**) Schematic representation of the workflow used in the comparison of embryonic and postnatal transcriptomes. The brachial domain (C4–T1) of *Mnx1-GFP* (in green) and *Chat^{IRESCre}::Ai9* (in red) mice was microdissected. Brachial GFP⁺ (at e12.5, scale bar: 20 μm) and tdTomato⁺ (at p8, scale bar: 100 μm) MNs were fluorescence-activated cell sorted and processed for RNA-sequencing. Spinal cord is outlined with white dashed line. (**B**) MA plot of differentially expressed genes. Green and

*Figure 1 continued on next page*

*Figure 1 continued*

red dots represent individual genes that are significantly (p<0.05) expressed (fourfold and/or higher) in embryonic and postnatal MNs, respectively. (C) Graphs showing fold enrichment for genes involved in specific biological processes. (D) Gene onthology analysis comparing protein class categories of highly expressed genes in embryonic (e12.5) and postnatal (p8) MNs. Green and red bars represent embryonic and postnatal genes, respectively.

The online version of this article includes the following figure supplement(s) for figure 1:

**Figure supplement 1.** Testing the specificity of genetic labeling of brachial motor neurons (MNs) at p8.

**Figure supplement 2.** RNA ISH analysis of terminal differentiation markers in *Hoxc8* MNΔ [early] mice.

combinations of TFs solely dedicated to either initiation or maintenance. Alternatively, initiation and maintenance can be achieved through the activity of the same, continuously expressed TF (or combinations thereof). Recent invertebrate and vertebrate studies on various neuron types support the latter mechanism (*Deneris and Hobert, 2014*; *Hobert and Kratsios, 2019*). We therefore sought to identify TFs with continuous expression, from embryonic to postnatal stages, in mouse brachial MNs.

First, we examined whether TFs from our embryonic (e12) RNA-Seq dataset continue to be expressed at postnatal stages (*Figure 1D*). We initially focused on 14 TFs from various families (e.g.

**Table 1.** Summary of candidate and unbiased approaches to reveal Hoxc8 target genes in mouse brachial MNs.

| | Gene name | Expression in WT brachial MNs | | | | Hoxc8 dependency | |
|---|---|---|---|---|---|---|---|
| | | e12 | p8 | p56 Allen Brain ISH | p60 snRNA-Seq dataset | *Hoxc8* MNΔ [early] *mice* | *Hoxc8* MNΔ [late] *mice* |
| | *Slc10a4* | + | + | + | + | No | N.D |
| | *Nrg1* | + | + | + | + | Yes | Yes |
| | *Nyap2* | + | + | N.D | + | No | N.D |
| | *Sncg* | + | + | + | + | No | N.D |
| | *Ngfr* | + | + | + | – | No | N.D |
| | *Glra2* | + | + | + | + | No | Yes |
| | *Cldn1* | N.D | – | N.D | – | N.D | N.D |
| Candidate approach | *Cacna1g* | N.D | – | + | + | N.D | N.D |
| | *Slc44a5* | + | + | + | + | No | N.D |
| | *Mcam* | + | + | + | + | Yes | Yes |
| | *Pappa* | + | + | + | + | Yes | Yes |
| | *Sema5a* | + | + | N.D | + | Yes | N.D |
| | *Pex14* | + | + | N.D | + | No | N.D |
| | *Tagln2* | + | + | + | – | No | N.D |
| | *Cldn19* | N.D | – | – | – | N.D | N.D |
| | *Wwc2* | N.D | – | + | + | N.D | N.D |
| | *Septin1* | N.D | – | N.D | N.D | N.D | N.D |
| | *Irx2* | + | + | + | – | N.D | N.D |
| | *Irx5* | + | + | + | – | N.D | N.D |
| RNA-Seq approach | *Irx6* | + | + | + | – | N.D | N.D |
| | *Ret* | + | + | N.D | + | Yes | No |
| Known Hoxc8 targets | *Gfra3* | + | – | – | – | Yes | N.D |

Expression in p56 brachial MNs was determined using the Allen Brain Map (http://portal.brain-map.org). We also interrogated the single nucleus (sn) RNA-seq datasets of p60 spinal MNs from http://spinalcordatlas.org/.

N.D: Not determined; + denotes expression; – denotes no expression.

RNA-Seq: RNA-sequencing.

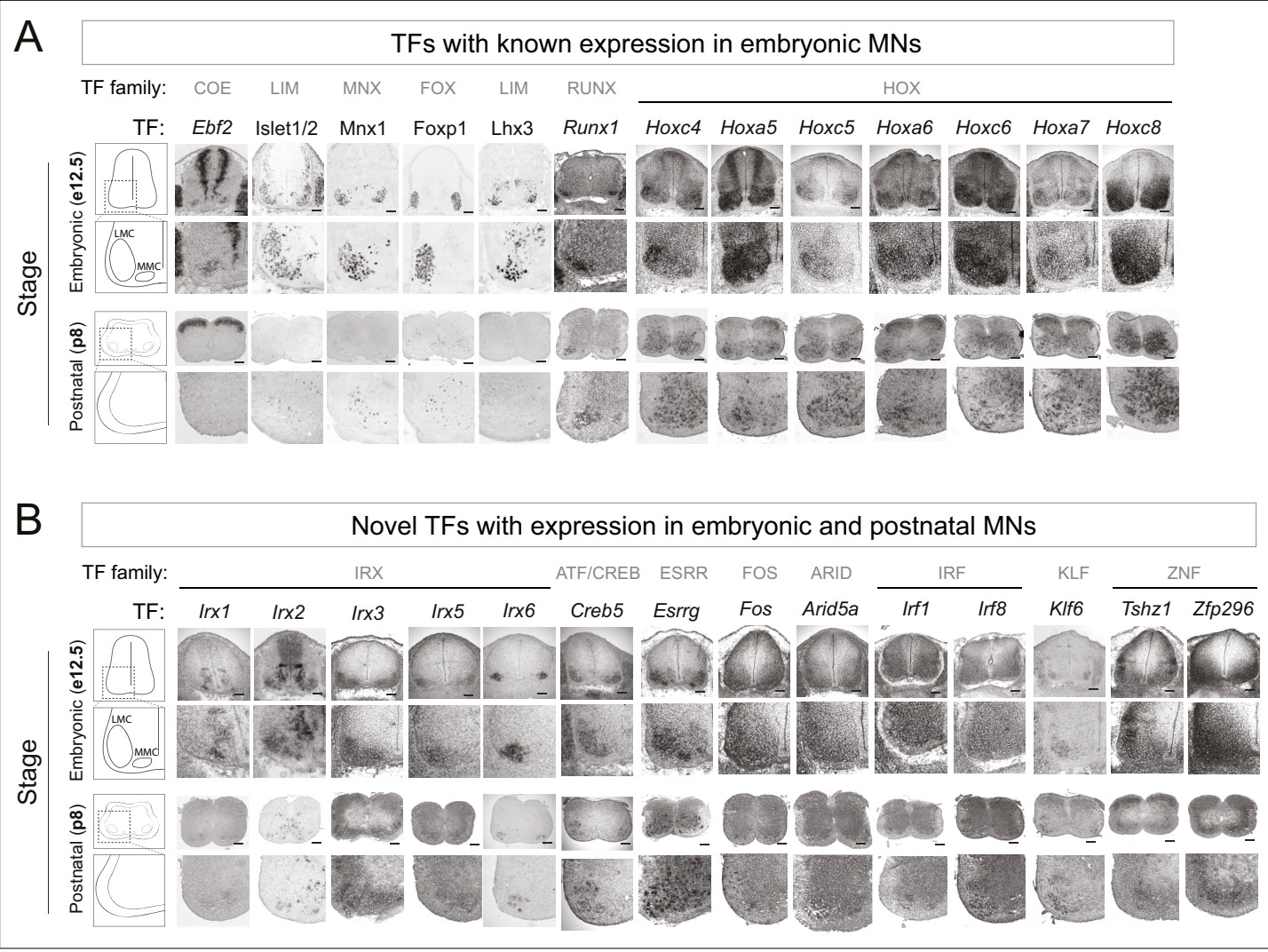

**Figure 2.** Known and novel transcription factors (TFs) are continuously expressed in brachial motor neurons (MNs) during embryonic and postnatal stages. (**A**) The expression of TFs with previously published roles in MN development was assessed in embryonic (e12.5) and postnatal (p8) spinal cords (N = 4) with RNA ISH (*Ebf2, Runx1, Hoxc4, Hoxa5, Hoxc5, Hoxa6, Hoxc6, Hoxa7, Hoxc8*) and immunohistochemistry (Islet1/2, Mnx1 [Hb9], Lhx3, Foxp1). Zoomed area of one side of the ventral spinal cord is shown below each image. (**B**) The expression of novel TFs was assessed in embryonic (e12.5) and postnatal (p8) spinal cords with RNA ISH (N = 4). Scale bar for e12.5 images: 50 μm; scale bar for p8 images: 250 μm.

LIM, Hox) with previously known embryonic expression and function in brachial MNs (*Ebf2, Islet1, Islet2, Hb9, Foxp1, Lhx3, Runx1, Hoxc4, Hoxa5, Hoxc5, Hoxa6, Hoxc6, Hoxa7, Hoxc8*) (*Arber et al., 1999*; *Catela et al., 2019*; *Catela et al., 2016*; *Dasen et al., 2008*; *Ericson et al., 1992*; *Philippidou and Dasen, 2013*; *Sharma et al., 1998*; *Stifani et al., 2008*; *Thaler et al., 1999*; *Thaler et al., 2004*; *Thaler et al., 2002*; *Tsuchida et al., 1994*). Through RNA ISH or antibody staining, we detected robust expression in brachial MNs at e12 for all 14 factors. Notably, 13 of these TFs continue to be expressed albeit at lower levels - in brachial MNs at p8 (*Figure 2A*, *Table 2*), suggesting these proteins - in addition to their known roles during early MN development - may exert other functions at later developmental and/or postnatal stages. Seven of these 13 proteins are TFs of the Hox family (*Hoxc4, Hoxa5, Hoxc5, Hoxa6, Hoxc6, Hoxa7, Hoxc8*) known to be expressed in brachial MNs at embryonic stages (*Philippidou and Dasen, 2013*), confirming the regional specificity of our RNA-Seq approach (*Figure 2A*). Moreover, our strategy is sensitive as it captured TFs with known expression in small populations of brachial MNs (e.g. MMC neurons), such as *Ebf2* and *Lhx3* (*Figure 2A*; *Catela et al., 2019*; *Sharma et al., 1998*).

**Table 2.** Validation of transcription factor expression in brachial MNs.

| TF | Type | Novel TF with MN expression | e12 MNs | p8 MNs | p56 MNs ISH Allen Brain | p60 snRNA-Seq dataset | Expression in other spinal cells at e12 |
|---|---|---|---|---|---|---|---|
| Ebf2 | Ebf/COE | No | + | – | – | – | + |
| Islet1 | LIM HD | No | + | + | + | N.D | + |
| Islet2 | LIM HD | No | + | + | + | N.D | + |
| Hb9 | HD | No | + | + | + | N.D | + |
| Foxp1 | FOX | No | + | + | – | + | – |
| Lhx3 | LIM HD | No | + | + | N.D | – | + |
| Runx1 | RUNX | No | + | + | + | – | – |
| Hoxc4 | HOX | No | + | + | + | + | + |
| Hoxa5 | HOX | No | + | + | N.D | – | + |
| Hoxc5 | HOX | No | + | + | + | + | + |
| Hoxa6 | HOX | No | + | + | – | – | + |
| Hoxc6 | HOX | No | + | + | N.D | – | + |
| Hoxa7 | HOX | No | + | + | + | – | + |
| Hoxc8 | HOX | No | + | + | N.D | – | + |
| Irx1 | IRO HD | Yes | + | + | + | – | + |
| Irx2 | IRO HD | Yes | + | + | + | – | + |
| Irx3 | IRO HD | Yes | + | + | + | – | – |
| Irx5 | IRO HD | Yes | + | + | + | – | – |
| Irx6 | IRO HD | Yes | + | + | + | – | – |
| Creb5 | CRE | Yes | + | + | + | + | – |
| Esrrg | NHR | Yes | + | + | – | + | + |
| Fos | FOS | Yes | + | + | + | – | + |
| Arid5a | ARID | Yes | + | + | – | – | + |
| Irf1 | IRF | Yes | + | + | – | – | + |
| Irf8 | IRF | Yes | + | + | – | – | + |
| Klf6 | KLF | Yes | + | + | – | + | – |
| Tshz1 | C2H2 Zn | Yes | + | + | + | + | + |
| Zfp296 | ZFP | Yes | + | + | + | – | + |
| Neurod6 | bHLH | N.A | – | – | N.D | – | Dorsal interneurons |
| Arid5b | ARID | N.A | – | – | N.D | + | Dorsal interneurons |
| Pou3f3 | POU | N.A | – | – | N.D | + | Dorsal interneurons |
| Mafb | bZIP | N.A | – | – | N.D | N.D | Ventral interneurons |
| Zfhx4 | Zn HD | N.A | – | – | N.D | + | Ventral interneurons |
| Elk3 | ETS | N.A | – | – | N.D | + | Vasculature |
| Epas1 | HIF | N.A | – | – | N.D | – | Vasculature |
| Heyl | bHLH | N.A | – | – | N.D | – | Vasculature |

Expression in p60 brachial MNs was determined using the Allen Brain Map (http://portal.brain-map.org). We also interrogated the single nucleus (sn) RNA-seq datasets of p60 spinal MNs from http://spinalcordatlas.org/. + denotes expression; – denotes no expression; N. D: Not determined; N. A: Not applicable.
RNA-Seq: RNA-sequencing.

We next sought to identify novel TFs with maintained expression in brachial MNs. We arbitrarily selected 22 genes from different TF families (15 TFs from the e12 dataset [*Irx1, Irx2, Irx3, Irx5, Irx6, Creb5, Esrrg, Neurod6, Arid5b, Pou3f3, MafB, Zfhx4, Elk3, Epas1, Heyl*] and 7 TFs from the p8 dataset [*Fos, Arid5a, Irf1, Irf8, Klf6, Tshz1, Zfp296*]). We detected persistent expression for 14 of these TFs in the embryonic (e12) and early postnatal (p8) brachial spinal cord. Expression was evident at the ventrolateral region, which is populated by MNs (*Figure 2B*, *Table 2*).

In conclusion, the expression of 13 TFs, with known roles in early MN development (e.g. cell specification, motor circuit assembly), is persistent at early postnatal stages (p8). Moreover, we identified 14 novel TFs from different families with expression in embryonic and postnatal (p8) brachial MNs (*Figure 2B*, *Table 2*). The continuous expression of all these factors suggests they may exert various functions in postmitotic MNs at different life stages. Consistent with this notion, some of these TFs are also expressed at later postnatal (p56, p60) stages in brachial MNs (*Table 2*).

## Hoxc8 controls expression of several terminal differentiation genes in e12 brachial MNs

In mice, Hox genes play critical roles during the early steps of spinal cord development, such as MN specification and circuit assembly (*Dasen et al., 2008*; *Dasen et al., 2003*; *Dasen et al., 2005*; *Philippidou and Dasen, 2013*). We found that several Hox genes (*Hoxc4, Hoxa5, Hoxc5, Hoxa6, Hoxc6, Hoxa7, Hoxc8*) are continuously expressed - from embryonic to postnatal stages - in brachial MNs (*Figure 2A*), but their function during later stages of MN development is largely unknown. This pattern of continuous Hox gene expression is reminiscent of recent observations in *C. elegans* nerve cord MNs (*Feng et al., 2020*; *Kratsios et al., 2017*). Importantly, *C. elegans* Hox genes are required not only to establish but also maintain at later stages the expression of multiple terminal differentiation genes (e.g. NT receptors, ion channels, signaling molecules) in nerve cord MNs (*Feng et al., 2020*).

Motivated by these findings in *C. elegans*, we sought to test the hypothesis that, in mice, Hox proteins control expression of terminal differentiation genes in spinal MNs. We focused on Hoxc8 because it is expressed in the majority of brachial MNs (segments C6-T2) (*Figure 2A*; *Catela et al., 2016*). Hoxc8 is not required for the overall organization of brachial MNs into columns, but - during early development (e12) - it controls forelimb muscle innervation by regulating *Gfrα3* and *Ret* expression in brachial MNs (*Catela et al., 2016*). However, whether Hoxc8 is involved in additional processes, such as the control of MN terminal differentiation, remains unclear.

To test this, we removed *Hoxc8* gene activity in brachial MNs. Because *Hoxc8* is also expressed in other spinal neurons (*Baek et al., 2019*; *Shin et al., 2020*; *Figure 2A*), we crossed *Hoxc8* [fl/fl] mice to *Olig2* [Cre] mice that enable *Cre* recombinase expression specifically in MN progenitors (*Figure 3A*; *Zawadzka et al., 2010*). This genetic strategy effectively removed Hoxc8 protein from postmitotic brachial MNs by e12 (*Figure 3B*). Because e12 is an early stage of MN differentiation (postmitotic MNs are generated between e9 and e11) (*Sims and Vaughn, 1979*), we will refer to the *Olig2* [Cre]*::Hoxc8* [fl/fl] mice as *Hoxc8* MNΔ [early]. Of note, the total number of brachial MNs (Mnx1+[HB9+] Isl1/2+) is unaffected in these animals at e12 (*Figure 3C*).

To test whether Hoxc8 controls expression of terminal differentiation genes, we initially followed a candidate approach. At e12, spinal MNs begin to acquire their terminal differentiation features, evident by the induction of genes coding for acetylcholine (ACh) biosynthesis proteins *(Slc18a3 [VAChT], Slc5a7[ChT1])* (*Martinez et al., 2012*). Consistently, *Slc18a3* and *Slc5a7* transcripts were captured in our e12 RNA-Seq dataset (*Figure 1D*). However, *Slc18a3* and *Slc5a7* expression was not affected in brachial MNs of *Hoxc8 MNΔ* [early] mice (*Figure 1—figure supplement 2*). Next, we tested the six newly identified terminal differentiation markers (*Slc10a4, Nrg1, Nyap2, Sncg, Ngfr, Glra2*) summarized in *Table 1*. We found that expression of *Neuregulin 1* (*Nrg1*), a molecule required for neuromuscular synapse maintenance and neurotransmission (*Mei and Xiong, 2008*; *Wolpowitz et al., 2000*), is reduced (but not eliminated) in e12 brachial MNs of *Hoxc8* MNΔ [early] mice (*Figure 3F*), likely due to the existence of additional factors that partially compensate for loss of *Hoxc8* gene activity. However, expression of the remaining five genes was unaffected in these animals (*Figure 1—figure supplement 2*), prompting us to devise an unbiased strategy to identify Hoxc8 targets.

We performed RNA-Seq on FACS-sorted brachial MNs from *Hoxc8* MNΔ [early]*::Mnx1-GFP* and control mice at e12 (see Materials and methods). We found dozens of significantly (p<0.05) upregulated (55)

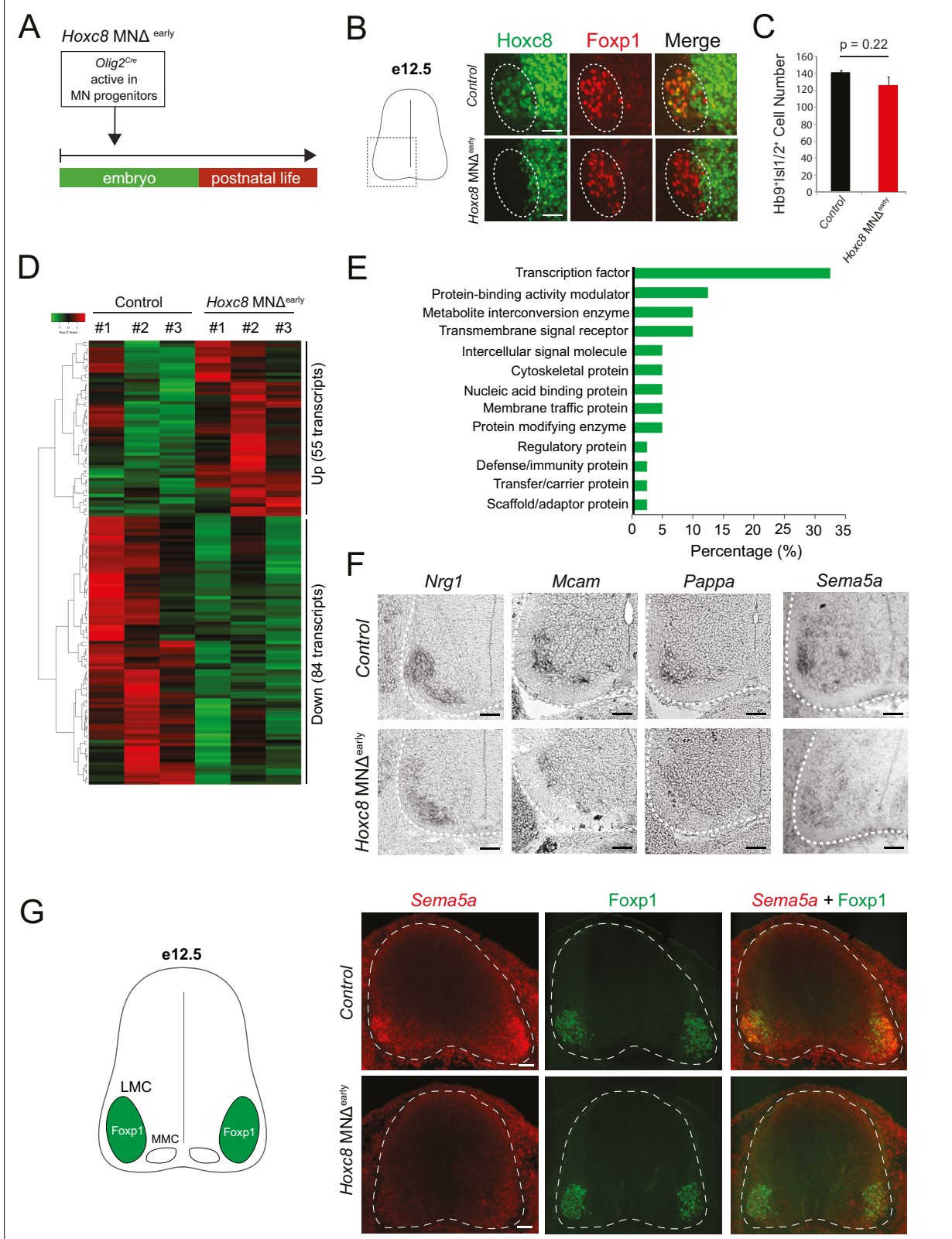

**Figure 3.** Early *Hoxc8* gene inactivation in brachial motor neurons (MNs) affects the expression of terminal differentiation genes. (**A**) Diagram illustrating genetic approach for *Hoxc8* gene inactivation during early MN development (*Hoxc8* MNΔ ᵉᵃʳˡʸ mice). (**B**) Immunohistochemistry showing that Hoxc8 protein (green) is not detected in Foxp1⁺ MNs (red, indicated with dashed ellipse) of *Hoxc8* MNΔᵉᵃʳˡʸ spinal cords at e12.5. Images of one side of the spinal cord are shown (boxed region in schematic at left). Scale bar: 50 μm. (**C**) Quantification of Mnx1⁺(Hb9⁺) Isl1/2⁺ MNs in e12.5 brachial spinal

*Figure 3 continued on next page*

*Figure 3 continued*

cords of *Hoxc8* MNΔ[early] and control (*Hoxc8*[fl/fl]) embryos (N = 4). (D) Heatmap showing upregulated and downregulated genes detected by RNA-Seq in control (*Hoxc8* [fl/fl]) and *Hoxc8* MNΔ[early] e12.5 MNs. Green and red colors, respectively, represent lower and higher gene expression levels. (E) Graphical percentage (%) representation of protein classes of the downregulated genes in *Hoxc8* MNΔ[early] spinal cords. (F) RNA ISH showing downregulation of *Nrg1, Mcam, Pappa,* and *Sema5a* mRNAs in brachial MNs of e12.5 *Hoxc8* MNΔ[early] spinal cords (N = 4). Spinal cord is outlined with a white dotted line. Scale bar: 50 μm. (G) RNA FISH for *Sema5a* coupled with antibody staining against Foxp1 (LMC marker) shows reduced *Sema5a* mRNA expression in Foxp1 +MNs of e12.5 *Hoxc8* MNΔ[early] spinal cords (N = 4). Images of a cross-section of the entire e12.5 spinal cord are shown. Scale bar: 40 μm.

The online version of this article includes the following figure supplement(s) for figure 3:

**Figure supplement 1.** RNA FISH analysis of terminal differentiation markers in *Hoxc8* MNΔ [early] mice.

and downregulated (84) transcripts in MNs lacking *Hoxc8* (*Figure 3D*). To test the hypothesis of *Hoxc8* being necessary to activate expression of MN terminal differentiation genes, we specifically focused on the list of 84 downregulated transcripts, which included two known Hoxc8 target genes (*Ret, Gfrα3*) (*Catela et al., 2016*) and *Hoxc8* itself (*Supplementary file 3*). GO analysis (see Materials and methods) on these 84 transcripts identified several putative Hoxc8 target genes encoding proteins from various classes (*Figure 3E*, *Supplementary file 3*). We focused on ion channels, transmembrane proteins, cell adhesion, and signaling molecules, as these constitute putative terminal differentiation markers (*Hobert, 2008*; *Hobert, 2011*). We selected nine genes (*Slc44a5, Mcam, Pappa, Sema5a, Pex14, Tagln2, Cldn19, Wwc2, Septin1*) and evaluated their expression with RNA ISH in brachial MNs at different stages. Five of these genes (*Slc44a5, Mcam, Pappa, Pex14, Tagln2*) are continuously expressed in brachial MNs at embryonic and postnatal stages (*Table 1*, *Figure 1—figure supplement 2*). Importantly, RNA ISH showed that expression of *Mcam*, a transmembrane cell adhesion molecule of the Immunoglobulin superfamily (*Gu et al., 2015*; *Taira et al., 2004*), and *Pappa*, a secreted molecule involved in skeletal muscle development (*Rehage et al., 2007*), is reduced at e12 in brachial MNs of *Hoxc8* MNΔ [early] mice (*Figure 3F–G*, *Figure 1—figure supplement 2*). Similar results for *Mcam* and *Pappa* were obtained with an RNA FISH method (*Figure 3—figure supplement 1*). In addition, we observed that *Sema5a* is expressed in embryonic (e12) but not postnatal brachial MNs, and this embryonic expression depends on Hoxc8 (*Table 1*, *Figure 3F–G*). Because *Sema5a* encodes a transmembrane protein of the Semaphorin protein family involved in axon guidance (*Duan et al., 2014*; *Hilario et al., 2009*; *Lin et al., 2009*), its dependency on *Hoxc8* could, at least partially, account for the previously reported MN axonal defects of *Hoxc8 MNΔ* [early] mice (*Catela et al., 2016*).

Altogether, this analysis identified 11 terminal differentiation genes with continuous expression in brachial MNs (*Slc10a4, Nrg1, Nyap2, Sncg, Ngfr, Glra2, Slc44a5, Mcam, Pappa, Pex14, Tagln2*), 3 of which (*Nrg1, Mcam, Pappa*) constitute Hoxc8 targets (*Table 1*). Although additional, yet-to-be identified TFs (potential Hoxc8 collaborators) must regulate the remaining eight genes, our findings do suggest Hoxc8 is involved in MN terminal differentiation. This new role for Hox in vertebrate MN development is consistent with recent studies in the *C. elegans* nerve cord, where Hox genes also control MN terminal differentiation (*Feng et al., 2020*; *Kratsios et al., 2017*).

## Hoxc8 is required to maintain expression of terminal differentiation genes in brachial MNs

Our analysis of Hoxc8 MNΔ [early] mice at e12 suggests Hoxc8 controls the early expression of select terminal differentiation genes (*Nrg1, Mcam, Pappa*) in brachial MNs. However, the persistent expression of *Hoxc8* both in embryonic and early postnatal MNs raises the intriguing possibility of a continuous requirement (*Figures 2A–4B*, *Figure 4—figure supplement 1*). Is *Hoxc8* required at later stages to maintain expression of terminal differentiation genes and thereby ensure the functionality of brachial MNs?

To address this, we crossed the *Hoxc8*[fl/fl] mice with the *Chat*[IRESCre] mouse line, which enables efficient gene inactivation in postmitotic MNs around e13.5–e14.5 (*Philippidou et al., 2012*; *Figure 4A*). Given that postmitotic MNs are generated between e9.5 and e11.5 (*Sims and Vaughn, 1979*), this genetic strategy preserves *Hoxc8* expression in MNs at least for 2 days after their generation. Consistent with a previous study that used this *Chat*[IRESCre] line (*Philippidou et al., 2012*), we observed Hoxc8 protein depletion in brachial MNs at e14.5 and later stages (*Figure 4B*, *Figure 4—figure supplement 1*). We will therefore refer to the *Chat*[IRESCre]*::Hoxc8*[fl/fl] animals as *Hoxc8 MNΔ* [late] because *Hoxc8* depletion in MNs occurs later compared to *Hoxc8 MNΔ* [early] mice (*Figure 4A*). Interestingly, expression of

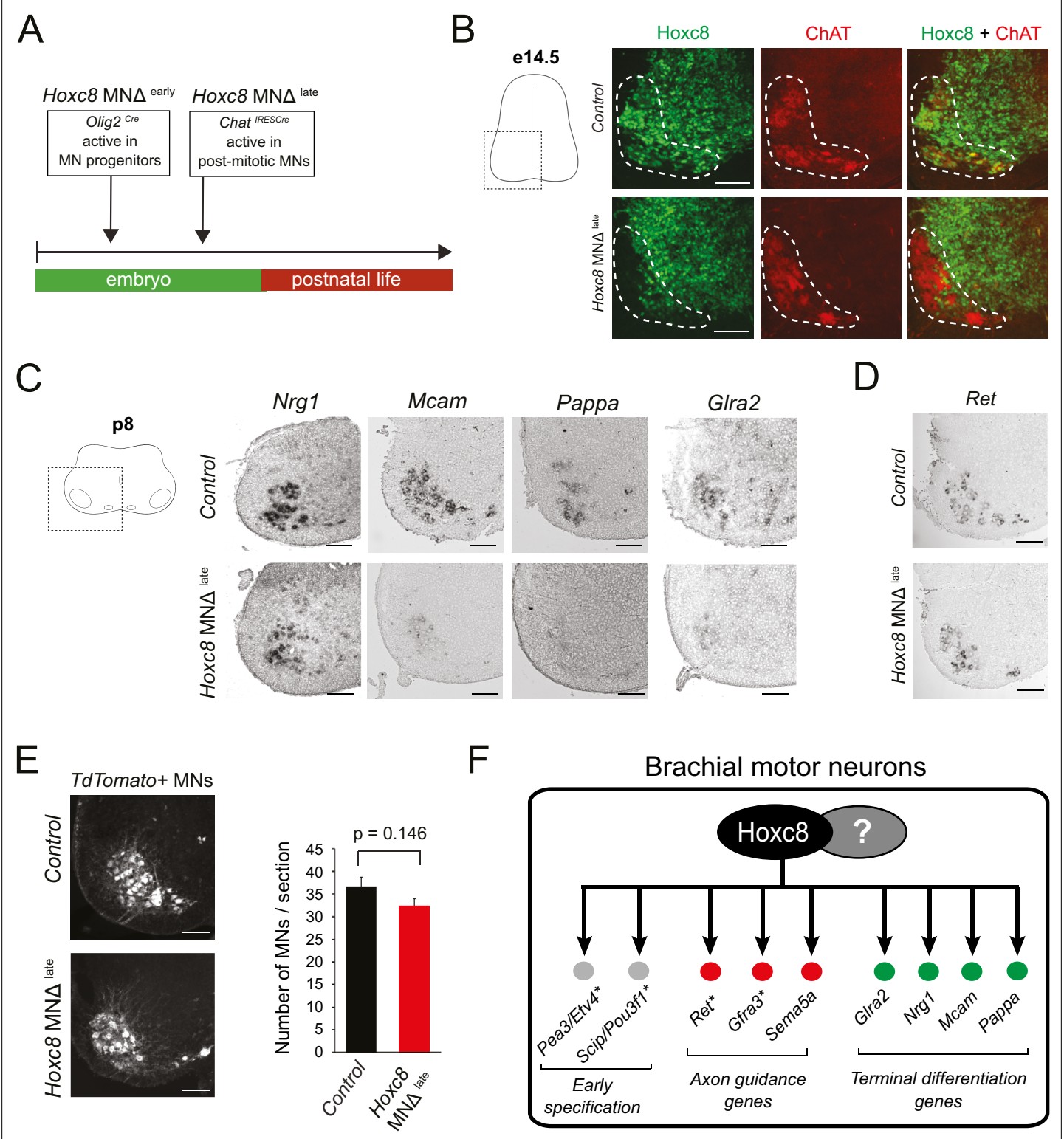

**Figure 4.** Late *Hoxc8* gene inactivation in brachial motor neurons (MNs) affects expression of terminal differentiation genes. (**A**) Diagram illustrating genetic approach for *Hoxc8* gene inactivation during late MN development. *Hoxc8* conditional mice were crossed with the *Chat^IRESCre* mouse line (*Hoxc8* MNΔ^late). (**B**) Immunohistochemistry showing that Hoxc8 protein (green) is not detected in ChAT-exprressing MNs (red) of *Hoxc8* MNΔ ^late spinal cords at e14.5 (N = 4). MN location is indicated with white dashed line. Hoxc8 is also expressed in other cell types outside the MN territory. Images of one side of the spinal cord are shown (boxed region in schematic at left). Scale bar: 100 μm. (**C**) RNA ISH showing reduced expression of *Pappa, Mcam, Glra2,* and

*Figure 4 continued on next page*

*Figure 4 continued*

*Nrg1* in *Hoxc8* MNΔ^late spinal cords at p8 (N = 4). Scale bar: 200 μm. (**D**) *Ret* expression is comparable between control and *Hoxc8* MNΔ^late spinal cords at p8 (N = 4). Scale bar: 200 μm. (**E**) Representative images and quantification of TdTomato-labeled MNs in p8 control (*Hoxc8^fl/fl::Ai9*) and *Hoxc8 MNΔ*^late (*Hoxc8^fl/fl::Chat^IRESCre::Ai9*) spinal cords (N = 4). Scale bar: 200 μm. (**F**) Schematic summarizing *Hoxc8* target genes in brachial MNs. Asterisks indicate previously known Hoxc8 target genes.

The online version of this article includes the following figure supplement(s) for figure 4:

**Figure supplement 1.** Depletion of Hoxc8 in brachial motor neurons (MNs) of *Hoxc8* MNΔ ^late mice.

the same terminal differentiation genes (*Nrg1, Mcam, Pappa*) we found affected in *Hoxc8* MNΔ ^early mice is also reduced in brachial MNs of *Hoxc8* MNΔ ^late mice at p8 (*Figure 4C*). This reduction is not due to secondary events affecting MN generation or survival because similar numbers of brachial MNs were observed in control and *Hoxc8* MNΔ ^late spinal cords at p8 (*Figure 4E*). Taken together, our findings on *Hoxc8* MNΔ ^early and *Hoxc8* MNΔ ^late mice strongly suggest a continuous requirement - *Hoxc8* is required to establish and maintain at later developmental stages the expression of several terminal differentiation genes in brachial MNs (*Figure 4F*).

## In brachial MNs, Hoxc8 partially modifies the suite of its target genes across different life stages

In the context of *C. elegans* MNs, our previous work revealed 'temporal modularity' in TF function (*Li et al., 2020*). That is, the suite of target genes of a continuously expressed TF, in the same cell type (e.g. MNs), is partially modified across different life stages. Here, we provide evidence for temporal modularity in *Hoxc8* function. We found that the terminal differentiation gene coding for the *glycine receptor subunit alpha-2 (Glra2)* (*Young-Pearse et al., 2006*) is affected in brachial MNs of *Hoxc8 MNΔ* ^late mice at p8 (*Figure 4C*). No effect was observed in MNs of *Hoxc8* MNΔ ^early mice at e12 (*Figure 1—figure supplement 2*), indicating a selective *Hoxc8* requirement for maintenance of *Glra2*. Conversely, the expression of *Ret*, a known *Hoxc8* target gene involved in MN axon guidance (*Bonanomi et al., 2012*), is selectively reduced in brachial MNs of *Hoxc8* MNΔ ^early animals at e12 (*Catela et al., 2016*), but remains unaffected in *Hoxc8 MNΔ* ^late animals at p8 (*Figure 4D*), suggesting Hoxc8 is only required for early *Ret* expression. Lastly, Hoxc8 can only activate expression of *Sema5a* (member of Semaphorin family) at embryonic stages (*Figure 3F–G*, *Table 1*). Contrary to these stage-specific Hoxc8 dependencies (Hoxc8 controls *Ret* and *Sema5a* at e12 and *Glra2* at p8), we also found that Hoxc8 is continuously required (both at e12 and p8) for expression of several terminal differentiation genes (*Nrg1, Mcam, Pappa*) (*Figures 3F and 4C*).

Altogether, these findings suggest that, in brachial MNs, Hoxc8 modifies the suite of its target genes at different developmental stages (*Figure 4F*). In Discussion, we elaborate on the functional significance of this phenomenon (temporal modularity).

## Hoxc8 is sufficient to induce its target genes and acts directly

To gain mechanistic insights, we analyzed recently published RNA-Seq and chromatin immunoprecipitation-sequencing (ChIP-seq) datasets on MNs derived from mouse embryonic stem cells (ESC), in which *Hoxc8* expression was induced with doxycycline (*Bulajić et al., 2020*). Our RNA-Seq analysis showed that induction of Hoxc8 (iHox8) resulted in upregulation of previously known (*Ret, Pou3f1 [Scip]*) and new (*Pappa, Glra2, Sema5a*) Hoxc8 target genes (*Figure 5A*). Moreover, ChIP-Seq for Hoxc8 in the context of these iHoxc8 ESC-derived MNs revealed binding in the *cis*-regulatory region of all these genes (*Figure 5B*), suggesting Hoxc8 acts directly to activate their expression. This in vitro data together with the in vivo findings in *Hoxc8* MNΔ ^early and *Hoxc8* MNΔ ^late mice (*Figure 3F–G*, *Figure 3—figure supplement 1*, *Figure 4C*) suggest that Hoxc8 is both necessary and sufficient for the expression of several of its target genes in spinal MNs.

Importantly, not all Hoxc8 target genes (e.g. *Nrg1, Mcam*) we identified in vivo are upregulated in iHoxc8 ESC-derived MNs (*Figure 5—figure supplement 1*). This is likely due to the lack of Hoxc8 collaborating factors in these in vitro generated MNs. A putative collaborator is Hoxc6 because (a) *Hoxc6* and *Hoxc8* are coexpressed in embryonic brachial MNs (*Catela et al., 2016*), (b) animals lacking either *Hoxc6* or *Hoxc8* in brachial MNs display similar axon guidance defects (*Catela et al., 2016*), and (c) Hoxc6 and Hoxc8 control the expression of the same axon guidance molecule (*Ret*) in brachial MNs

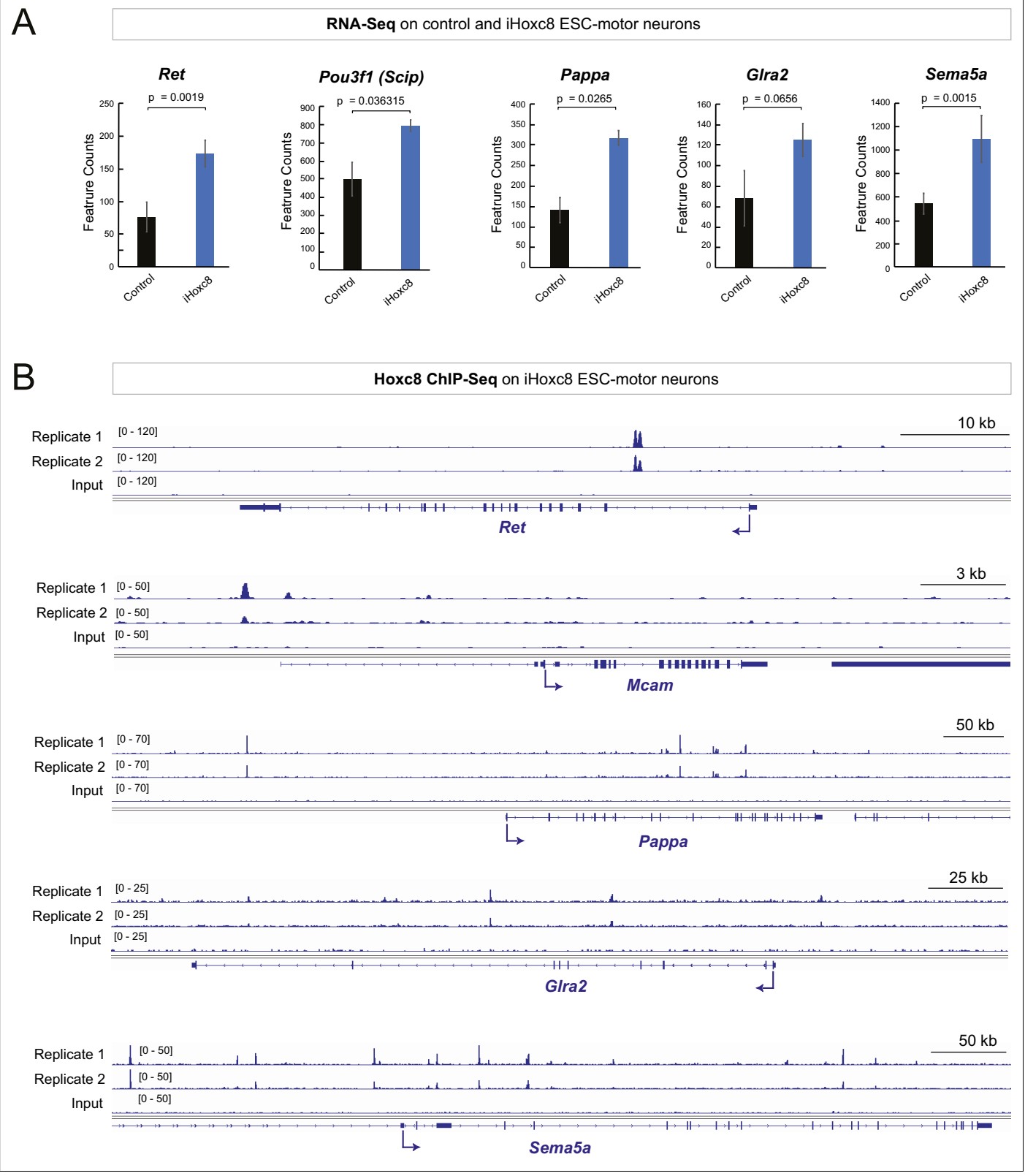

**Figure 5.** Hoxc8 sufficiency and direct mode of action. (**A**) Analysis of RNA-sequencing (RNA-Seq) data from control and iHoxc8 motor neurons (MNs) shows Hoxc8 is sufficient to induce the expression of previously known *(Ret, Pou3f1[Scip])* and new *(Pappa, Glra2, Sema5a)* Hoxc8 target genes. GEO accession numbers: Control (GSM4226469, GSM4226470, GSM4226471) and iHoxc8 (GSM4226475, GSM4226476, GSM4226477). (**B**) Analysis of chromatin immunoprecipitation-sequencing (ChIP-Seq) data from iHoxc8 MNs shows Hoxc8 directly binds to the *cis*-regulatory region of its target

*Figure 5 continued on next page*

*Figure 5 continued*

genes (*Ret, Mcam, Pappa, Glra2, Sema5a*). GEO accession numbers: Input (GSM4226461) and iHoxc8 replicates (GSM4226436, GSM4226437). Snapshots of each gene locus were generated with Integrative Genomics Viewer (IGV, Broad Institute).

The online version of this article includes the following figure supplement(s) for figure 5:

**Figure supplement 1.** Hoxc6 and Hoxc8 bind to the same *cis*-regulatory regions of Hoxc8 target genes.

(*Catela et al., 2016*). Supporting the notion of collaboration, our analysis of available ChIP-seq data for Hoxc6 and Hoxc8 from iHoxc6 and iHoxc8 ESC-derived MNs (*Bulajić et al., 2020*), respectively, showed that these Hox proteins bind directly on the *cis*-regulatory region of previously known (*Ret, Gfra3*) and new (*Mcam, Pappa, Nrg1, Sema5a*) Hoxc8 target genes (*Figure 5—figure supplement 1*).

## *Hoxc8* gene activity is necessary for brachial motor neuron function

We next sought to assess any potential behavioral defects in adult *Hoxc8* MNΔ $^{early}$ and *Hoxc8* MNΔ $^{late}$ animals by evaluating their motor coordination (*Deacon, 2013*), forelimb grip strength (*Takeshita et al., 2017*), and treadmill performance (*Wozniak et al., 2019*). No defects were observed in *Hoxc8* MNΔ $^{early}$ and *Hoxc8* MNΔ $^{late}$ mice during the rotarod performance test (*Figure 6—figure supplement 1*), suggesting balance and motor coordination are normal in these animals. Next, we evaluated forelimb grip strength because brachial MNs innervate forelimb muscles. We found a statistically significant defect in *Hoxc8* MNΔ$^{early}$ mice, but not in *Hoxc8* MNΔ$^{late}$ mice (*Figure 6A–B*). Lastly, we tested these animals for their ability to run on a treadmill for a period of 30 s. At a low speed (15 cm/s), we observed statistically significant defects in *Hoxc8* MNΔ$^{early}$ mice. That is, 64.28% of *Hoxc8* MNΔ$^{early}$ mice fell off the treadmill in the first 5 s of the trial compared to 28.57% of control mice (p=0.0108) (*Figure 6C, Figure 6—videos 1; 2*). Moreover, 0% of *Hoxc8* MNΔ$^{early}$ mice were able to stay longer than 20 s on the treadmill compared to 42.85% of control mice (*Figure 6C*). On the other hand, statistically significant defects were observed in *Hoxc8* MNΔ$^{late}$ mice only when the treadmill speed was increased to 25 cm/s (*Figure 6C–D*). That is, 43.33% of *Hoxc8* MNΔ $^{late}$ mice fell off the treadmill in the first 5 s of the trial compared to 17.39% of control mice (p=0.0461) (*Figure 6D, Figure 6—videos 3; 4*). Together, these data show that *Hoxc8* MNΔ$^{late}$ mice display a milder behavioral phenotype compared to *Hoxc8* MNΔ$^{early}$ mice. This is likely due to the fact that *Hoxc8* MNΔ$^{early}$ mice display a composite phenotype i.e. defects in early MN specification and axon guidance (*Catela et al., 2016*) combined with terminal differentiation defects (this study), whereas the *Hoxc8* MNΔ$^{late}$ mice only display terminal differentiation defects (this study). Although we cannot exclude the possibility that the terminal differentiation defects of *Hoxc8* MNΔ$^{early}$ mice are a consequence of their early MN specification defects, this is unlikely as Hoxc8 binds directly to the *cis*-regulatory region of terminal differentiation genes (*Mcam, Pappa, Glra2*) (*Figure 5B*).

## Hox gene expression is maintained in thoracic and lumbar MNs at postnatal stages

In brachial MNs, we found that the expression of multiple Hox genes (*Hoxc4, Hoxa5, Hoxc5, Hoxa6, Hoxc6, Hoxa7, Hoxc8*) is maintained from embryonic to early postnatal stages (*Figure 2A*). We wondered whether sustained Hox gene expression in MNs is a broadly applicable theme to other rostrocaudal domains of the spinal cord. We therefore performed RNA-Seq on thoracic and lumbar FACS-isolated MNs from *ChAT$^{IRESCre}$::Ai9* mice at p8 (see Materials and methods) (*Figure 6—figure supplement 2*). Our analysis indeed revealed that, similar to our observations in the brachial domain, additional Hox genes are expressed postnatally (p8) in thoracic (*Hoxd9*) and lumbar (*Hoxa10, Hoxc10, Hoxa11*) MNs (*Figure 6—figure supplement 2A-C*). We further confirmed these findings with RNA ISH (*Figure 6—figure supplement 2D*). While the functions of some of these Hox genes are known during the early steps of MN development (*Philippidou and Dasen, 2013*), their continuous expression suggests additional roles at later embryonic and postnatal stages. Genetic inactivation of these genes at successive developmental stages will determine whether they function in a manner similar to *Hoxc8*, suggesting a more general Hox-based strategy for the control of spinal MN terminal differentiation.

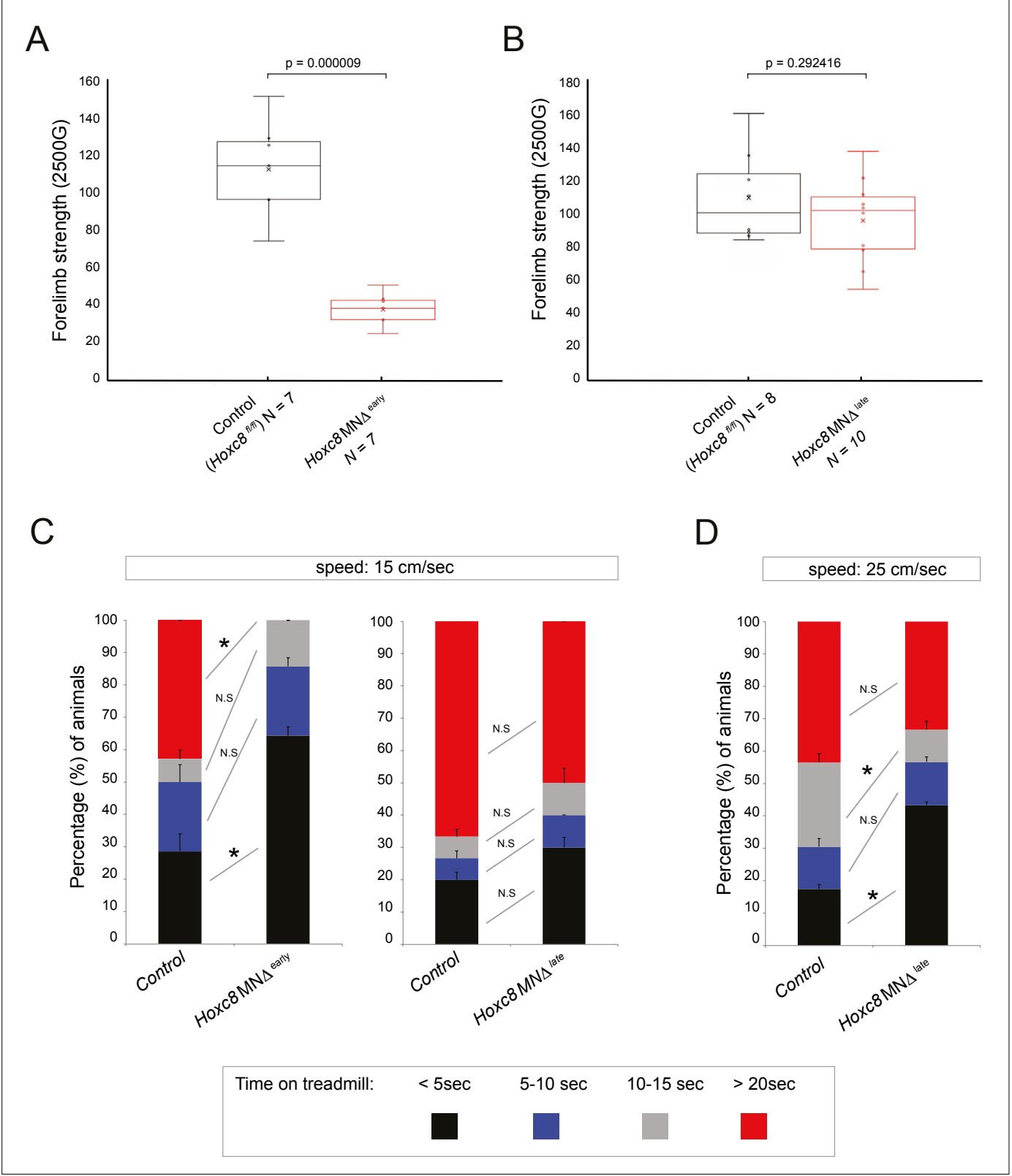

**Figure 6.** Brachial motor neuron (MN) function is impaired upon Hoxc8 depletion. (**A**) Forelimb grip strength analysis on control (*Hoxc8 <sup>fl/fl</sup>*, N = 7) and *Hoxc8* MNΔ <sup>early</sup> (N = 8) adult mice. See Methods for details. (**B**) Forelimb grip strength analysis on control (*Hoxc8 <sup>fl/fl</sup>*, N = 7) and *Hoxc8* MNΔ <sup>late</sup> (N = 8) adult mice. (**C**). Treadmill analysis (at 15 cm/s speed) on control (*Hoxc8 <sup>fl/fl</sup>*, N = 7) and *Hoxc8* MNΔ <sup>early</sup> (N = 8) adult mice, as well as on control (*Hoxc8 <sup>fl/fl</sup>*, N = 8) and *Hoxc8* MNΔ <sup>late</sup> (N = 10) adult mice. See Methods for details. Asterisk (*) indicates p=0.0108. Experiment repeated twice. (**D**). Treadmill

*Figure 6 continued on next page*

*Figure 6 continued*

analysis (at 25 cm/s speed) on control (*Hoxc8* $^{fl/fl}$, N = 8) and *Hoxc8* MNΔ $^{late}$ (N = 10) adult mice. Treadmill speed at 25 cm/s. Asterisk (*) indicates p=0.0461. Experiment repeated three times. The 30-s long videos were analyzed and data were binned into four categories based on the duration of each mouse's stay on the treadmill (category 1 [black]: <5 s; category 2 [blue]: 5–10 s; category 3 [gray]: 10–15 s; category 4 [red]: >20 s).

The online version of this article includes the following video and figure supplement(s) for figure 6:

**Figure supplement 1.** Rotarod performance test on *Hoxc8* MNΔ $^{early}$ and *Hoxc8* MNΔ $^{late}$ mice.

**Figure supplement 2.** Different Hox genes are expressed in brachial, thoracic, and lumbar motor neurons (MNs) at postnatal day 8.

**Figure 6—video 1.** Treadmill test on control (*Hoxc8* $^{fl/fl}$) littermate of *Hoxc8* MNΔ $^{early}$ mice.
https://elifesciences.org/articles/70766/figures#fig6video1

**Figure 6—video 2.** Treadmill test on a *Hoxc8* MNΔ $^{early}$ mouse.
https://elifesciences.org/articles/70766/figures#fig6video2

**Figure 6—video 3.** Treadmill test on control (*Hoxc8* $^{fl/fl}$) littermate of *Hoxc8* MNΔ $^{late}$ mice.
https://elifesciences.org/articles/70766/figures#fig6video3

**Figure 6—video 4.** Treadmill test on a *Hoxc8* MNΔ $^{late}$ mouse.
https://elifesciences.org/articles/70766/figures#fig6video4

## Discussion

Somatic MNs in the spinal cord innervate hundreds of skeletal muscles and control a variety of motor behaviors, such as locomotion, skilled movement, and postural control. Although we are beginning to understand the molecular programs that control the early steps of spinal MN development (*Osseward and Pfaff, 2019*, *Philippidou and Dasen, 2013*; *Stifani, 2014*), how these clinically relevant cells acquire and maintain their terminal differentiation features (e.g. NT phenotype, electrical, and signaling properties) remains poorly understood. In this study, we focused on the brachial region of the mouse spinal cord and determined the molecular profile of postmitotic MNs at a developmental and a postnatal stage. This longitudinal approach identified genes with continuous expression in brachial MNs, encoding novel TFs and effector molecules critical for neuronal terminal differentiation (e.g. ion channels, NT receptors, signaling proteins, adhesion molecules). Interestingly, we also found that most TFs, previously implicated in the early steps of brachial MN development (e.g. initial specification, axon guidance, circuit assembly), such as LIM- and Hox-type TFs (*di Sanguinetto et al., 2008*, *Philippidou and Dasen, 2013*; *Stifani, 2014*), continue to be expressed in these cells postnatally (p8). Such maintained expression suggested additional roles for these factors during later developmental stages. To test this idea, we focused on *Hoxc8*, identified its target genes, and uncovered a continuous requirement for Hoxc8 in the establishment and maintenance of select MN terminal differentiation features. Our findings dovetail recent Hox studies in the *C. elegans* nervous system (*Feng et al., 2020*; *Kratsios et al., 2017*; *Zheng et al., 2015*) and suggest an evolutionarily conserved role for Hox proteins in the control of neuronal terminal differentiation.

### Hoxc8 partially modifies the suite of its target genes to control multiple aspects of brachial MN development

Despite their fundamental roles in patterning the vertebrate hindbrain and spinal cord (*Krumlauf, 2016*; *Parker and Krumlauf, 2017*; *Philippidou and Dasen, 2013*), the downstream targets of Hox proteins in the nervous system remain poorly defined. In this study, we uncovered several *Hoxc8* target genes encoding different classes of proteins (*Sema5a* - axon guidance molecule; *Glra2, Nrg1, Mcam, Pappa* - terminal differentiation genes) (*Figures 3E and 4F*), suggesting *Hoxc8* controls different aspects of brachial MN development through the regulation of these genes.

In mice, Hoxc8 is expressed in MNs of the MMC and LMC columns between segments C6 and T1 of the spinal cord (*Catela et al., 2016*; *Tiret et al., 1998*), herein referred to as 'brachial MNs'. Previous studies using either global *Hoxc8* knock-out or *Hoxc8* MNΔ $^{early}$ mice reported aberrant connectivity of forelimb muscles (*Catela et al., 2016*; *Tiret et al., 1998*). It was proposed that this early developmental phenotype likely arises due to reduced expression of axon guidance molecules, such as *Ret* and *Gfrα3*, in brachial MNs of *Hoxc8* MNΔ $^{early}$ mice (*Catela et al., 2016*). Another early developmental defect previously observed in *Hoxc8* MNΔ $^{early}$ mice is the reduced expression of MN pool-specific markers (*Pou3f1 [Scip], Etv4 [Pea3]*) within the LMC (*Figure 4F*), albeit the overall organization of

brachial MNs into MMC and LMC columns appears normal (*Catela et al., 2016*). Although these findings implicate Hoxc8 in the early steps of brachial MN development, it remained unclear whether Hoxc8 controls additional aspects of MN development during later stages.

In this study, we propose that Hoxc8 controls select features of brachial MN terminal differentiation, such as the expression of the glycine receptor subunit *Glra2*, the cell adhesion molecule *Mcam*, the secreted signaling protein *Pappa*, and a molecule associated with neurotransmission and neuromuscular synapse maintenance (*Nrg1*). We found that all these molecules are expressed continuously in embryonic and postnatal (p8) brachial MNs. By removing *Hoxc8* gene activity either at an early (*Hoxc8* MNΔ $^{early}$ mice) or late (*Hoxc8* MNΔ $^{late}$ mice) developmental stage, we uncovered a continuous Hoxc8 requirement for the initial expression and maintenance of *Mcam*, *Pappa*, and *Nrg1*. Intriguingly, we also found evidence for temporal modularity in Hoxc8 function, that is, the suite of Hoxc8 targets in brachial MNs is partially modified at different developmental stages. Two lines of evidence support this notion: (a) expression of the terminal differentiation gene *Glra2* is only affected in *Hoxc8* MNΔ $^{late}$ mice, indicating a selective *Hoxc8* requirement for *Glra2* maintenance in MNs, and (b) expression of two axon guidance molecule (*Sema5a*, *Ret*) is only affected in MNs of *Hoxc8* MNΔ $^{early}$ mice.

What is the purpose of such temporal modularity? We propose that Hoxc8 partially modifies the suite of its target genes at different life stages to control different facets of brachial MN development, such as early MN specification, axon guidance, and terminal differentiation (*Figure 4F*). During early development, Hoxc8 controls early specification markers (*Etv4 [Pea3], Pou3f1[Scip]*), as well as axon guidance molecules, such as *Ret* (*Bonanomi et al., 2012*; *Catela et al., 2016*) and *Sema5a* (this study) in order to ensure proper MN-muscle connectivity. Consistent with this idea, similar axon guidance defects occur in *Hoxc8* and *Ret* mutant mice (*Catela et al., 2016*). During late development, Hoxc8 maintains the expression of the glycine receptor subunit *Glra2*, a terminal differentiation marker necessary for glycinergic input to brachial MNs (*Young-Pearse et al., 2006*). Apart from Hoxc8, temporal modularity has been recently described for two other TFs: UNC-3 in *C. elegans* MNs and Pet-1 in mouse serotonergic neurons (*Li et al., 2020*; *Wyler et al., 2016*). Like Hoxc8, UNC-3 and Pet-1 control various aspects of *C. elegans* motor and mouse serotonergic neurons (e.g. axon guidance, terminal differentiation) (*Donovan et al., 2019*; *Kratsios et al., 2011*, *Liu et al., 2010*; *Prasad et al., 1998*). Although the mechanistic basis of such modularity remains poorly understood, a possible scenario is the employment of transient enhancers – a mechanism recently proposed for maintenance of gene expression in in vitro differentiated spinal MNs (*Rhee et al., 2016*). We surmise that temporal modularity in TF function may be a broadly applicable mechanism enabling a single TF to control different, temporally segregated 'tasks/processes' within the same neuron type.

## A new role for Hox in the mouse nervous system: establishment and maintenance of neuronal terminal differentiation

Much of our current understanding of Hox protein function in the nervous system stems from studies in the vertebrate hindbrain and spinal cord, as well as the *Drosophila* ventral nerve cord (*Baek et al., 2013*; *Baek et al., 2019*; *Estacio-Gómez and Díaz-Benjumea, 2014*; *Estacio-Gómez et al., 2013*; *Karlsson et al., 2010*; *Mendelsohn et al., 2017*; *Miguel-Aliaga and Thor, 2004*; *Moris-Sanz et al., 2015*; *Parker and Krumlauf, 2017*; *Philippidou and Dasen, 2013*). This large body of work has established Hox proteins as critical regulators of the early steps of neuronal development including cell specification, migration, survival, axonal path finding, and circuit assembly. However, the functions of Hox proteins in later steps of nervous system development remain poorly understood. Recent work on invertebrate Hox genes has begun to address this knowledge gap. In *Drosophila* MNs necessary for feeding, *Deformed* (*Dfd*) is required to maintain neuromuscular synapses (*Friedrich et al., 2016*). In *C. elegans* touch receptor neurons, the anterior (*ceh-13*) and posterior (*egl-5*) Hox genes control the expression levels of the LIM homeodomain protein MEC-3, which in turn controls touch receptor terminal differentiation (*Zheng et al., 2015*). In the *C. elegans* ventral nerve cord, midbody (*lin-39*, *mab-5*) and posterior (*egl-5*) Hox genes control distinct terminal differentiation features of midbody and posterior MNs, respectively (*Kratsios et al., 2017*). LIN-39 binds to the *cis*-regulatory region of multiple terminal differentiation genes (e.g. ion channels, NT receptors, signaling molecules) and is required for their maintained expression in MNs during postembryonic stages (*Feng et al., 2020*).

Our Hoxc8 findings in mice support the hypothesis that Hox-mediated control of later aspects of neuronal development (e.g. terminal differentiation) is evolutionarily conserved from invertebrates to

mammals. Similar to *C. elegans* Hox genes, mouse *Hoxc8* is continuously expressed in brachial MNs from embryonic to early postnatal stages, and sustained *Hoxc8* gene activity is required to establish and maintain at later developmental stages the expression of several terminal differentiation genes. This noncanonical, late function of *Hoxc8* may be shared by other Hox genes in the mouse nervous system. In the spinal cord, we found several other Hox genes (*Hoxc4, Hoxa5, Hoxc5, Hoxa6, Hoxc6, Hoxa7*) to be continuously expressed in brachial MNs, potentially acting as Hoxc8 collaborators. We made similar observations in thoracic (*Hoxd9*) and lumbar (*Hoxc10, Hoxa11*) MNs (*Figure 6—figure supplement 2*). Moreover, expression of multiple Hox genes has been observed in the adult mouse and human brain, leading to the hypothesis that maintained Hox gene expression is necessary for activity-dependent synaptic pruning and maturation (*Hutlet et al., 2016*; *Takahashi et al., 2004*). To date, the functional significance of maintained Hox gene expression in the CNS remains largely unknown, and temporally controlled genetic approaches are required to fully elucidate the late functions of this remarkable class of highly conserved TFs.

## The quest for terminal selectors of spinal motor neuron identity

Numerous genetic studies in the nematode *C. elegans* support the idea that continuously expressed TFs (termed 'terminal selectors') establish during development and maintain throughout postembryonic life the identity and function of individual neuron types by activating the expression of terminal differentiation genes (e.g. NT biosynthesis components, ion channels, adhesion, and signaling molecules) (*Deneris and Hobert, 2014*; *Hobert, 2008*; *Hobert, 2016*). Multiple cases of terminal selectors for various neuron types have already been described in flies, cnidarians, marine chordates, and mice, suggesting deep conservation for this type of regulators (*Allan and Thor, 2015*; *Deneris and Hobert, 2014*; *Hobert, 2008*; *Hobert, 2016*; *Hobert and Kratsios, 2019*; *Tournière et al., 2020*). However, it remains unclear whether spinal MNs in vertebrates employ a terminal selector type of mechanism to acquire and maintain their terminal differentiation features. Addressing this knowledge gap could aid the development of in vitro protocols for the generation of mature and terminally differentiated spinal MNs, a much anticipated goal in the field of MN disease modeling (*Sances et al., 2016*).

Three lines of evidence implicate *Hoxc8* in the control of MN terminal differentiation. First, *Hoxc8* is expressed continuously, from embryonic to early postnatal stages, in brachial MNs. Second, our in vivo data and in vitro analysis suggest Hoxc8 is both necessary and sufficient for the expression of several of its target genes in MNs - such mode of action is reminiscent of terminal selectors (*Flames and Hobert, 2009*; *Kratsios et al., 2011*). Third, both early and late removal of *Hoxc8* in brachial MNs affected the expression of several terminal differentiation genes, suggesting a continuous requirement. However, *Hoxc8* does not act alone - loss of *Hoxc8* did not completely eliminate the expression of its target genes (*Figures 3F–G and 4C*). This residual expression indicates that additional TFs are necessary to control brachial MN terminal differentiation. As mentioned in Results, one such factor is *Hoxc6*, which is coexpressed with *Hoxc8* in brachial MNs during embryonic and postnatal stages (*Catela et al., 2016*; *Figure 2A*). Importantly, Hoxc6 and Hoxc8 bind directly on the *cis*-regulatory regions of the same terminal differentiation genes in the context of mouse ESC-derived MNs (*Figure 5—figure supplement 1*). Another putative Hoxc8 collaborator is the LIM homeodomain protein *Islet1* (*Isl1*), which is required for early induction of genes necessary for ACh biosynthesis in mouse spinal MNs and the in vitro generation of MNs from human pluripotent stem cells (*Cho et al., 2014*; *Qu et al., 2014*; *Rhee et al., 2016*). Interestingly, *Isl1* is expressed continuously in brachial MNs (*Figure 2*) and amplifies its own expression (*Erb et al., 2017*) - both defining features of a terminal selector gene. In addition to Hoxc6 and Isl1, our expression analysis revealed multiple TFs from different families (e.g. Hox, Irx, LIM) with continuous expression in brachial MNs (*Figure 2*, *Table 2*). In the future, temporally controlled gene inactivation studies are needed to determine whether these TFs participate in the control of spinal MN terminal differentiation. Intriguingly, the majority of the TFs with continuous expression in brachial MNs belong to the homeodomain family. Homeodomain TFs are overrepresented in the current list of *C. elegans* and mouse terminal selectors (*Deneris and Hobert, 2014*; *Reilly et al., 2020*; *Serrano-Saiz et al., 2013*), suggesting an ancient role for this family of regulatory factors in the control of neuronal terminal differentiation.

# Materials and methods

**Key resources table**

| Reagent type (species) or resource | Designation | Source or reference | Identifiers | Additional information |
|---|---|---|---|---|
| Genetic reagent (*Mus musculus*) | *Mnx1-GFP* | PMID:12176325 | Not available | Not available |
| Genetic reagent (*M. musculus*) | *Ai9* | PMID:20023653 | MGI: J:155,793 | Not available |
| Genetic reagent (*M. musculus*) | *Hoxc8 $^{fl/fl}$* | PMID:19621436 | Not available | Not available |
| Genetic reagent (*M. musculus*) | *Olig2$^{Cre}$* | PMID:18046410 | MGI: 3774124 | Not available |
| Genetic reagent (*M. musculus*) | *Chat$^{IRESCre}$* | PMID:21284986 | MGI: J:169,562 | Not available |
| Antibody | anti-ChAT (Goat polyclonal) | Millipore | Cat# AB144P, RRID:AB_2079751 | IF (1:100) |
| Antibody | anti-FoxP1 (Rabbit polyclonal) | Dasen lab | CU1025 | IF(1:32000) |
| Antibody | anti-RFP (Rabbit polyclonal) | Rockland | Cat# 600-401-379S, RRID:AB_11182807 | IF(1:500) |
| Antibody | anti-Alexa 488-Hoxc8 (mouse monoclonal) | Dasen lab | Not applicable | IF(1:1500) |
| Antibody | anti-GFAP (Chicken polyclonal) | Millipore | Cat# AB5541, RRID:AB_177521 | IF(1:200) |
| Antibody | anti-CD11b (Rat monoclonal) | Bio-Rad | Cat# MCA711, RRID:AB_321292 | IF(1:50) |
| Antibody | anti-mPea3 (Rabbit polyclonal) | Dasen lab | Not applicable | IF(1:32000) |
| Antibody | anti-Digoxigenin-POD, Fab fragments (Sheep polyclonal) | Roche Diagnostics Deutschland GmbH | Cat# 11207733910 | IF(1:3000) |
| Antibody | Cy3 AffiniPure anti-Goat IgG (Donkey polyclonal) | Jackson ImmunoResearch Labs | Cat# 705-165-147, RRID:AB_2307351 | IF(1:800) |
| Antibody | Alexa Fluor 488 anti-Rabbit IgG (Donkey) | Thermo Fisher Scientific | Cat# A-21206, RRID:AB_2535792 | IF(1:1000) |
| Antibody | Cy3 AffiniPure anti- Rabbit IgG (Donkey polyclonal) | Jackson ImmunoResearch Labs | Cat# 711-165-152, RRID:AB_2307443 | IF(1:800) |
| Antibody | Alexa Fluor 488 anti-Goat IgG (Donkey polyclonal) | Thermo Fisher Scientific | Cat# A-11055, RRID:AB_2534102 | IF(1:1000) |
| Antibody | Alexa Fluor 488 anti-mouse IgG (Donkey polyclonal) | Thermo Fisher Scientific | Cat# A-21202, RRID:AB_141607 | IF(1:1000) |
| Antibody | Alexa Fluor 488 anti-Chicken IgY (Goat polyclonal) | Thermo Fisher Scientific | Cat# A32931, RRID:AB_2762843 | IF(1:1000) |
| Antibody | Alexa Fluor 488 anti-Rat IgG (Goat polyclonal) | Thermo Fisher Scientific | Cat# A-11006, RRID:AB_2534074 | IF(1:1000) |
| Software, algorithm | ZEN | ZEISS | RRID: SCR_013672 | Version 2.3.69.1000, Blue edition |
| Software, algorithm | Fiji | Image J | RRID: SCR_003070 | Version 1.52i |

## Mouse husbandry and genetics

All mouse procedures were approved by the Institutional Animal Care and Use Committee (IACUC) of the University of Chicago (Protocol No. 72463). The generation of *Hoxc8 $^{floxed/floxed}$* (*Blackburn et al., 2009*), *Olig2$^{Cre}$* (*Dessaud et al., 2007*), *Mnx1-GFP* (*Wichterle et al., 2002*), *ChAT-IRES-Cre* (*Rossi et al., 2011*), and *Ai9* (*Madisen et al., 2010*) mice has been previously described. Mendelian ratios at weaning stage for *Hoxc8* MNΔ $^{early}$ and Hoxc8 MNΔ $^{late}$ animals are provided in *Supplementary file 4*.

## Fluorescence-activated cell sorting and RNA-Seq of brachial motor neurons

For the analysis shown in *Figure 1*, spinal cord segments C4-T1 of e12.5 *Mnx1-GFP* and p8 *Chat^IRES-Cre^::Ai9* animals were microdissected using the dorsal root ganglia as reference. For the analysis shown in *Figure 3*, segments C7-T2 were used. The spinal cord tissue was dissociated using papain and filtered (using 50 µm filters) for sorting. A GFP negative spinal cord was also included as a negative control for the FACS setup. DAPI staining was used to exclude dead cells from the sorting. FACS-sorted MNs were collected into Arcturus Picopure extraction buffer and immediately processed for RNA isolation. RNA was extracted from purified MNs, using the Arcturus Picopure RNA isolation kit (Arcturus, #KIT0204). For the RNA-Seq analysis on e12.5 *Mnx1-GFP* embryos, three biological replicates were used; five to six spinal cords were pooled per replicate. For the RNA-Seq analysis on p8 *Chat^IRESCre^::Ai9* animals, three biological replicates were used; three spinal cords were pooled per replicate. RNA quality and quantity were measured with an Agilent Picochip (Agilent 2100 Bioanalyzer). All samples had high quality scores between 9 and 10 RIN. After cDNA library preparation, RNA-Seq was performed using an Illumina HiSeq 4000 sequencer (50-nucleotide single-end reads, University of Chicago Genomics Core facility).

## RNA-Seq analysis

Raw sequence data were subjected to quality control using the FastQC algorithm (http://www.bioinformatics.babraham.ac.uk/projects/fastqc/). Unique reads were aligned into the mouse genome (GRCm38/mm10) using the HISET2 alignment program *Kim et al., 2015* followed by transcript counting with the featureCounts program (*Liao et al., 2014*). Differential gene expression analysis was performed with the DESeq2 program (*Love et al., 2014*). All analyses were performed using the open source, web-based Galaxy platform (https://usegalaxy.org). The heatmaps were generated using the Morpheus program developed by the Broad Institute (https://software.broadinstitute.org/morpheus). Gene hierarchical clustering was performed using a Pearson's correlation calculation.

## RNA in situ hybridization

E12.5 embryos and p8 spinal cords were fixed in 4% paraformaldehyde for 1.5–2 hr and overnight, respectively, placed in 30% sucrose overnight (4 °C) and embedded in optimal cutting temperature compound. Cryosections were generated and processed for ISH or immunohistochemistry as previously described (*Dasen et al., 2005*; *De Marco Garcia and Jessell, 2008*).

## Fluorescent RNA ISH coupled with antibody staining

Cryosections were postfixed in 4% paraformaldehyde, washed in PBS, endogenous peroxidase was blocked with a 0.1% $H_2O_2$ solution and permeabilized in PBS/0.1% Triton-X100. Upon hybridization with DIG-labeled RNA probe overnight at 72°C and washes in SSC, the anti-DIG antibody conjugated with peroxidase (Roche) and primary antibody against Foxp1 (rabbit anti-Foxp1, Dr. Jeremy Dasen) were applied overnight (4 °C) to the sections. The next day, the sections were incubated with the secondary antibody (Alexa 488 donkey anti-rabbit IgG, Life Technologies, A21206), and detection of RNA was performed using a Cy3 Tyramide Amplification system (Perkin Elmer). Images were obtained with a high-power fluorescent microscope (Zeiss Imager V2) and analyzed with Fiji software (*Schindelin et al., 2012*).

## Immunohistochemistry

Fluorescence staining on cryosections was performed as previously described (*Catela et al., 2016*).

## Gene ontology analysis

Protein classification was performed using the Panther Classification System Version 15.0 (http://www.pantherdb.org). Embryonic (1381 out of 2904) and postnatal (1348 out of 2699) MN genes were categorized into protein classes using the algorithms built into Panther (*Mi et al., 2013*; *Thomas et al., 2003*).

## Rotarod performance test

Female mice were trained on an accelerating rotarod for 5 days. The experimenter was blind to the genotypes. For the *Hoxc8* MNΔ^early^ analysis, seven control (*Hoxc8^fl/fl^*) and seven (*Olig2^Cre^::Hoxc8^fl/fl^*)

mice were used at the age of 4–5 months. For the *Hoxc8* MNΔ $^{late}$ analysis, 8 control (*Hoxc8$^{fl/fl}$*) and 10 (*Chat$^{IRESCre}$::Hoxc8$^{fl/fl}$*) mice were used at the age of 2–5 months. A computer-controlled rotarod apparatus (Rotamex-5, Columbus Instruments, Columbus, OH, USA) with a rat rod (7-cm diameter) was set to accelerate from 4 to 40 revolutions per minute (rpm) over 300 s, and recorded time to fall. Mice received five consecutive trials per session, one session per day (about 60 s between trials).

### Forelimb grip strength test

The forelimb strength of female mice was measured using a grip strength meter from Bioseb (model BIO-GS3). For the *Hoxc8* MNΔ $^{early}$ analysis, seven control (*Hoxc8$^{fl/fl}$*) and seven (*Olig2$^{Cre}$::Hoxc8$^{fl/fl}$*) mice were used at the age of 4–5 months. For the *Hoxc8* MNΔ $^{late}$ analysis, 8 control (*Hoxc8$^{fl/fl}$*) and 10 (*Chat$^{IRESCre}$::Hoxc8$^{fl/fl}$*) mice were used at the age of 2–5 months. We followed the manufacturer's protocol. In brief, the meter was positioned horizontally on a heavy metal shelf (provided by the manufacturer), assembled with a grip grid. Mice were held by the tail and lowered toward the apparatus. The mice were allowed to grasp the metal grid only with their forelimbs and were then pulled backward in the horizontal plane. The maximum force of grip was measured, and we used the average of six measurements for analysis. Force was measured in Newton and Grams. The experimenter was blind to the genotypes.

### Treadmill test

The treadmill test was conducted on female mice by using the DigiGait system (MouseSpecifics, Inc), which is equipped with a motorized transparent treadmill belt and a high-speed digital camera that provides images of the ventral side of the mouse (*Figure 6—videos 1–4*). For the *Hoxc8* MNΔ $^{early}$ analysis, seven control (*Hoxc8$^{fl/fl}$*) and seven (*Olig2$^{Cre}$::Hoxc8$^{fl/fl fl}$*) mice at the age of 4–5 months were placed onto the walking compartment. The treadmill was turned on at a speed of 15 cm/s. For the *Hoxc8* MNΔ $^{late}$ analysis, 8 control (*Hoxc8$^{fl/fl}$*) and 10 (*Chat$^{IRESCre}$::Hoxc8$^{fl/fl}$*) mice at the age of 2–5 months were placed onto the walking compartment. The treadmill test was conducted at two different speeds (15 cm/s and 25 cm/s). The 30-s long videos were obtained for each mouse. Videos were analyzed and data were binned into four categories based on the duration of each mouse's stay on the treadmill (category 1: <5 s; category 2: 5–10 s; category 3: 10–15 s; category 4: >20 s).

### Statistical analysis

For data quantification, graphs show values expressed as mean ± SEM. With the exception of the rotarod and treadmill experiments, all other statistical analyses were performed using the unpaired *t*-test (two-tailed). Differences with p<0.05 were considered significant. For the rotarod performance test, two-way ANOVA was performed (Prism Software). For the treadmill experiment, we used Fisher's exact test.

## Acknowledgements

We are grateful to members of the Kratsios lab (Yinan Li, Edgar Correa, Nidhi Sharma, Filipe Goncalves Marques) and Drs. Deeptha Vasudevan, Ellie Heckscher, and Oliver Hobert for comments on the manuscript. We thank Dr. Jeremy Dasen (NYU) for providing the following antibodies (rabbit anti-Foxp1, rabbit anti-Lhx3, rabbit anti-Hb9, rabbit anti-Isl1/2), Milica Bulajić for help obtaining the RNA-Seq and ChIP-Seq data on in vitro differentiated iHoxc6 and iHoxc8 motor neurons, and Jihad Aburas for technical assistance. We thank the following Core Facilities at The University of Chicago: (a) Cytometry and Antibody Technology, and (b) Genomics Facility (RRID:SCR019196), especially Dr. Pieter Faber, for his assistance with the RNA-Sequencing. This work was supported by the Lohengrin Foundation and a grant from the National Institute of Neurological Disorders and Stroke (NINDS) of the NIH (Award Number: R01NS116365) to PK. This publication was also supported by a grant from the Robert Packard Center for ALS Research at Johns Hopkins University. Its contents are solely the responsibility of the authors and do not necessarily represent the official views of The Johns Hopkins University or any grantor providing funds to its Robert Packard Center for ALS Research.

# Additional information

## Funding

| Funder | Grant reference number | Author |
|---|---|---|
| National Institute of Neurological Disorders and Stroke | R01NS116365 | Paschalis Kratsios |
| Robert Packard Center for ALS Research, Johns Hopkins University | Not applicable | Paschalis Kratsios |
| Lohengrin Foundation | Not applicable | Paschalis Kratsios |

The funders had no role in study design, data collection and interpretation, or the decision to submit the work for publication.

## Author contributions

Catarina Catela, Conceptualization, Data curation, Formal analysis, Investigation, Methodology, Visualization, Writing - original draft, Writing - review and editing; Yihan Chen, Yifei Weng, Kailong Wen, Formal analysis, Investigation, Validation; Paschalis Kratsios, Conceptualization, Funding acquisition, Investigation, Project administration, Supervision, Visualization, Writing - original draft, Writing - review and editing

## Author ORCIDs

Paschalis Kratsios (iD) http://orcid.org/0000-0002-1363-9271

## Ethics

This study was performed in strict accordance with the recommendations in the Guide for the Care and Use of Laboratory Animals of the National Institutes of Health. All of the animals were handled according to approved institutional animal care and use committee (IACUC) protocol (#72463) of the University of Chicago.

## Decision letter and Author response

Decision letter https://doi.org/10.7554/eLife.70766.sa1
Author response https://doi.org/10.7554/eLife.70766.sa2

---

# Additional files

## Supplementary files

• Supplementary file 1. List of enriched transcripts in embryonic (e12.5) and postnatal (p8) brachial motor neurons.

• Supplementary file 2. Gene ontology (GO) analysis on enriched transcripts from embryonic (e12.5) and postnatal (p8) brachial motor neurons.

• Supplementary file 3. List of downregulated and up-regulated transcripts in brachial motor neurons of *Hoxc8Δ* $^{early}$ mice at e12.5.

• Supplementary file 4. Mendelian ratios at weaning stage for *Hoxc8* MNΔ $^{early}$ and *Hoxc8* MNΔ $^{late}$ animals.

• Transparent reporting form

## Data availability

Sequencing data have been deposited in GEO under accession code GSE174709. All data generated or analyzed in this study are included in the manuscript and supporting files.

The following dataset was generated:

| Author(s) | Year | Dataset title | Dataset URL | Database and Identifier |
|---|---|---|---|---|
| Catela C, Kratsios P | 2021 | New roles for Hoxc8 in the establishment and maintenance of motor neuron identity | https://www.ncbi.nlm.nih.gov/geo/query/acc.cgi?acc=GSE174709 | NCBI Gene Expression Omnibus, GSE174709 |

The following previously published dataset was used:

| Author(s) | Year | Dataset title | Dataset URL | Database and Identifier |
|---|---|---|---|---|
| Mahony S | 2020 | Diversification of posterior Hox patterning by graded ability to engage inaccessible chromatin | https://www.ncbi.nlm.nih.gov/geo/query/acc.cgi?acc=GSE142379 | NCBI Gene Expression Omnibus, GSE142379 |

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
