## [Editor Report]

This manuscript will be of interest to developmental geneticists interested in neuroscience, and how spinal motor neurons maintain their unique identities in adulthood after fate decisions are made in the embryo. The work here demonstrates that a Hox transcription factor acts as a terminal selector to control motor neuron identity, thus mirroring recent studies in *C. elegans*, and thus pointing towards this type of gene regulation as important in building diverse nervous systems.

---

## [Decision Letter]

**Decision letter after peer review:**

Thank you for submitting your article "Control of spinal motor neuron terminal differentiation through sustained Hoxc8 gene activity" for consideration by *eLife*. Your article has been reviewed by 3 peer reviewers, one of whom is a member of our Board of Reviewing Editors, and the evaluation has been overseen by Piali Sengupta as the Senior Editor. The following individual involved in review of your submission has agreed to reveal their identity: Aaron D Gitler (Reviewer #2).

Essential revisions:

The reviewers are in agreement that this is a very interesting study, addressing a fundamental topic in developmental genetics, with important implications for the field. Nonetheless, the reviewers have coalesced around several items that need to be addressed before we can consider this paper for publication.

1. The behavioral data is potentially important to this study, but not robustly performed. The N numbers need to be increased and additional behavioral tests should be added. Moreover, deleting Hoxc8 expression earlier in these studies, as was done for other experiments, would be useful.

2. The quality of many of the RNA in situ experiments was not good, making it hard for reviewers to easily assess the data. The authors should repeat the in situs where necessary or even consider newer, more quantitative RNA in situ approaches.

3. Authors should clarify their conclusions about the role of Hoxc8 in brachial motor neuron differentiation because at least from the data presented it does not seem to have a very significant role, and the authors should consider the potential involvement of other collaborating factors.

4. Authors should clarify the potential caveats by using two different methods for labeling MNs at the two time points and possibly coming up with a better method or way to normalize the data.

*Reviewer #1 (Recommendations for the authors):*

1. My major concern with this paper is the lack of specificity for the findings and ease in building a cohesive story. The paper is largely centered around RNAseq and RNA in situ (that is sometimes hard to interpret, see comment below). I appreciate the large amount of effort put into this study, but it would be nice to know how many of these gene regulatory networks are direct (ie. TF binding to consensus motifs in promoter regions). I was unable to follow a clear logic as to why certain genes were chosen for follow up, or to focus large parts of the story on.

2. In regards to Figure 1, how specific are HB9::GFP and Chat::Ai9 to motor neurons, or neurons in general. Authors should co-stain for other neuronal and glial markers. Also, why not use a reporter from one of the identified terminal differentiation factors which is on early and late, so that a single reporter is used?

3. The RNA in situs in Figures3A, 4E, 5B and some of the supplemental images are not of the highest quality and one has to read the manuscript to know what to conclude. Perhaps newer approaches like RNAscope, or similar probes, would be more quantitative and easier to interpret. Also some images have white dotted circles around them cells of interest and other images do not – please be consistent.

4. Although sympathetic to the need to use the Hoxc8 late mice for behavioral experiments due to perinatal lethality with the Hoxc8 early mice, these experiments demonstrate there is no phenotype and the author's assertion that mutant mice "tend" to perform worse is an unfair stretch. Moreover, with an N number of 3 and 5 mice on some of the experiments, its hard to make conclusions. The authors should consider consulting with or collaborating with a more behavior focused lab to do these experiments in a more robust manner, adding more tests and mice, or consider leaving the behavior experiments out of this paper.

*Reviewer #2 (Recommendations for the authors):*

We have the following comments and suggestions for the authors to consider:

1) Hox factors are known to control motor neuron subtype/pool specification (e.g. Dasen et al., 2005). It would be helpful to know if Hoxc8 early or late deletions change motor neuron subtype/pool identity rather than their maturation, which could be the reason for the differential gene expression This could be confirmed with co-labeling of reporter and subtype/pool markers, as in the Irx experiments.

2) Based on Figure 4F, the authors seem to distinguish terminal differentiation genes (e.g. neurotransmitters, synapse molecules) from axon guidance molecules and transcription factors. However, there are times when these seem to be treated interchangeably (e.g. discussion of Sema5a and Ret on page 12). The authors should more clearly and consistently state what they consider terminal differentiation markers in the text.

3) It would be worthwhile to have co-labeling with reporter (Hb9-GFP vs ChAT-tdTom) for at least some of the selected genes (e.g. in Figure 3G) to see if there is specific reduction in the cells of interest rather than just through correlation based on spatial pattern.

4) The Irx2 knockout experiments suggest that this downstream target of Hoxc8 may be involved in specification of limb-generating motor neurons, but not in the expression of terminal differentiation markers regulated by Hoxc8. This result seems tangential to the paper as 1) the knockouts are performed very early (prior to cell specification) and therefore does not provide any additional information about how Hoxc8 regulates motor neuron development past specification stage and 2) do not a clear functional link between Hoxc8 and terminal differentiation regulation.

5) It is unclear what the *C. elegans* Irx2 (fosmid) experiments add to this paper. The loss of MNs entirely seems to detract from the overall point of this paper, which is that Hoxc8 helps to establish/maintain brachial motor neuron identity/maturation.

6) It is unclear what the behavioral experiments add to the paper since would be very difficult to attribute any phenotype to a specific cause. Also, unclear why the experiments were performed only in late Hoxc8 deletion and not the early deletion as well.

7) The quality of some of the in situs should be improved (e.g. Glra2 in Figure 4E)

8) It might be of interest to overexpress Hoxc8 in other regions of the spinal cord (e.g. cervical or thoracic) using chick electroporation for example to see if it can lead to overexpression of some of these terminal differentiation genes. This gain-of-function experiment may lend additional support to the proposal that Hoxc8 is required to maintain expression of these genes, although I am not sure that this is a requirement for distinguishing terminal selectors.

*Reviewer #3 (Recommendations for the authors):*

1. The authors used two different genetic systems to label brachial MNs at embryonic day 12 (e12) and postnatal day 8 (p8) as it was not possible for them to label MNs with just a single genetic system. Using these reagents they conclude that Hoxc8 regulates some of the same and some different targets. But can they rule out that some of these results is a consequence of using two different labeling systems? Are they certain that the cells labeled at both time points are the same cells? Perhaps using a lineage tracing tool and/or normalizing with some of the genes they discover in their RNA-seq experiments may be a way add confidence in the similarities and differences that they find in the RNAseq datasets are accurate.

2. Along the same lines, the RNA ISH of Hoxc8 at e12 looks dense, suggesting that all MNs in LMC and MMC regions express Hoxc8. On the other hand, expression of Hoxc8 seems sparse at p8, suggesting few Hoxc8+ MNs at p8 compared to e12 stage. Performing dual RNA ISH with Hoxc8 and endogenous genes of interest would provide greater confidence that the same number of Hoxc8+ MNs are present at two different time points.

3. The authors state that Hoxc8 MN late mutants perform worse in rotarod performance test and forelimb grip strength in 3-month old mice. However, the statistical analysis suggests there is no significant change. It was unclear why the authors chose such a late time point for these assays, when the mice potentially have time to compensate for a compromised motor system. Redoing this experiment at earlier time points, such as with 1 month old mice (there is precedence for this in the literature), might reveal significant differences.

---

## [Author Response]

Essential revisions:The reviewers are in agreement that this is a very interesting study, addressing a fundamental topic in developmental genetics, with important implications for the field. Nonetheless, the reviewers have coalesced around several items that need to be addressed before we can consider this paper for publication.1. The behavioral data is potentially important to this study, but not robustly performed. The N numbers need to be increased and additional behavioral tests should be added. Moreover, deleting Hoxc8 expression earlier in these studies, as was done for other experiments, would be useful.

We completely agree. In the revised manuscript, we have included new data on three behavioral tests (forelimb grip strength, treadmill, rotarod). We increased the number of animals (N) and also included in the analysis two sets of mice with conditional Hoxc8 manipulations: 1) *Hoxc8 MNΔ^early^* mice in which *Olig2-Cre* is active in motor neuron (MN) progenitors, and 2) *Hoxc8 MNΔ^late^* mice in which ChAT-IRES-Cre is active in post-mitotic MNs.

We have added the following paragraph in Results that summarizes our findings:

“Hoxc8 gene activity is necessary for brachial motor neuron function

We next sought to assess any potential behavioral defects in adult Hoxc8 MNΔ ^early^ and Hoxc8 MNΔ ^late^ animals by evaluating their motor coordination (Deacon, 2013), forelimb grip strength (Takeshita et al., 2017) and treadmill performance(Wozniak et al., 2019). No defects were observed in Hoxc8 MNΔ ^early^ and Hoxc8 MNΔ ^late^ mice during the rotarod performance test (Figure 6 —figure supplement 1), suggesting balance and motor coordination are normal in these animals. Next, we evaluated forelimb grip strength because brachial MNs innervate forelimb muscles. We found a statistically significant defect in Hoxc8 MNΔ^early^ mice, but not in Hoxc8 MNΔ^late^ mice (Figure 6A-B). Lastly, we tested these animals for their ability to run on a treadmill for a period of 30 seconds (sec). At a low speed (15 cm/sec), we observed statistically significant defects in Hoxc8 MNΔ^early^ mice. That is, 64.28% of Hoxc8 MNΔ^early^ mice fell off the treadmill in the first 5 sec of the trial compared to 28.57% of control mice (p = 0.0108) (Figure 6C, Figure 6 – videos 1-2). Moreover, 0% of Hoxc8 MNΔ^early^ mice were able to stay longer than 20 sec on the treadmill compared to 42.85% of control mice (Figure 6C). On the other hand, statistically significant defects were observed in Hoxc8 MNΔ^late^ mice only when the treadmill speed was increased to 25 cm/sec (Figure 6C-D). That is, 43.33% of Hoxc8 MNΔ ^late^ mice fell off the treadmill in the first 5 sec of the trial compared to 17.39% of control mice (p = 0.0461) (Figure 6D, Figure 6 – videos 3-4). Together, these data show that Hoxc8 MNΔ^late^ mice display a milder behavioral phenotype compared to Hoxc8 MNΔ^early^ mice. This is likely due to the fact that Hoxc8 MNΔ^early^ mice display a composite phenotype i.e., defects in early MN specification and axon guidance (Catela et al., 2016) combined with terminal differentiation defects [this study], whereas the Hoxc8 MNΔ^late^ mice only display terminal differentiation defects (this study). Although we cannot exclude the possibility that the terminal differentiation defects of MNΔ^early^ mice are a consequence of their early MN specification defects, this is unlikely as Hoxc8 binds directly to the cis-regulatory region of terminal differentiation genes (Mcam, Pappa, Glra2) (Figure 5B).”

2. The quality of many of the RNA in situ experiments was not good, making it hard for reviewers to easily assess the data. The authors should repeat the in situs where necessary or even consider newer, more quantitative RNA in situ approaches.

We conducted new RNA fluorescent ISH (FISH) experiments coupled with a MN-specific marker (Foxp1). These new data corroborate our original conclusion: Hoxc8 controls the expression of several target genes in brachial MNs (new Figure 3G, new Figure 3 —figure supplement 1). To complement these in vivo findings, we now provide evidence in in vitro generated MNs that Hoxc8 is sufficient to induce the expression of the same target genes (e.g., *Pappa, Glra2, Sema5a*) we identified in vivo (new Figure 5A).

3. Authors should clarify their conclusions about the role of Hoxc8 in brachial motor neuron differentiation because at least from the data presented it does not seem to have a very significant role, and the authors should consider the potential involvement of other collaborating factors.

We completely agree. Our new molecular analysis and behavioral data suggest Hoxc8 does not act alone in the context of brachial motor neuron terminal differentiation. We have clarified our conclusions and further highlighted the involvement of other collaborating factors in multiple places in Results and Discussion. The most significant changes to address this important point are summarized below.

In Results:

“We found that expression of Neuregulin 1 (Nrg1), a molecule required for neuromuscular synapse maintenance and neurotransmission(Mei and Xiong, 2008, Wolpowitz et al., 2000), is reduced (but not eliminated) in e12 brachial MNs of Hoxc8 MNΔ ^early^ mice (Figure 3F), likely due to the existence of additional factors that partially compensate for loss of Hoxc8 gene activity.”

“Importantly, not all Hoxc8 target genes (e.g., Nrg1, Mcam) we identified in vivo are upregulated in iHoxc8 ESC-derived MNs (Figure 5 —figure supplement 1). This is likely due to the lack of Hoxc8 collaborating factors in these in vitro generated MNs. A putative collaborator is Hoxc6 because (a) Hoxc6 and Hoxc8 are co-expressed in embryonic brachial MNs (Catela et al., 2016), (b) animals lacking either Hoxc6 or Hoxc8 in brachial MNs display similar axon guidance defects (Catela et al., 2016), and (c) Hoxc6 and Hoxc8 control the expression of the same axon guidance molecule (Ret) in brachial MNs (Catela et al., 2016). Supporting the notion of collaboration, our analysis of available ChIP-seq data for Hoxc6 and Hoxc8 from iHoxc6 and iHoxc8 ESC-derived MNs(Bulajic et al., 2020), respectively, showed that these Hox proteins bind directly on the cis-regulatory region of previously known (Ret, Gfra3) and new (Mcam, Pappa, Nrg1, Sema5a) Hoxc8 target genes (Figure 5 —figure supplement 1).”

In Discussion:

“In the spinal cord, we found several other Hox genes (Hoxc4, Hoxa5, Hoxc5, Hoxa6, Hoxc6, Hoxa7) to be continuously expressed in brachial MNs, potentially acting as Hoxc8 collaborators.”

“This residual expression indicates that additional TFs are necessary to control brachial MN terminal differentiation. As mentioned in Results, one such factor is Hoxc6, which is co-expressed with Hoxc8 in brachial MNs during embryonic and postnatal stages (Catela et al., 2016) (Figure 2A). Importantly, Hoxc6 and Hoxc8 bind directly on the cis-regulatory regions of the same terminal differentiation genes in the context of mouse ESC-derived MNs (Figure 5 —figure supplement 1). Another putative Hoxc8 collaborator is the LIM homeodomain protein Islet1 (Isl1), which is required for early induction of genes necessary for ACh biosynthesis in mouse spinal MNs and the in vitro generation of MNs from human pluripotent stem cells(Cho et al., 2014, Qu et al., 2014, Rhee et al., 2016).”

4. Authors should clarify the potential caveats by using two different methods for labeling MNs at the two time points and possibly coming up with a better method or way to normalize the data.

We completely agree. The potential caveat is that our FACS-sorted cells also include a minority of cells that are not motor neurons. We acknowledge this possibility in Results:

“Two factors that could contribute to these transcriptional differences between the e12 and p8 datasets are: (1) different levels of gene expression (see next Section), and (2) a small fraction of the FACS-sorted cells are not MNs. Indeed, Hb9 (Mnx1) and ChAT, in addition to MNs, are also expressed in small, non-overlapping neuronal populations in the spinal cord (Wilson et al., 2005, Zagoraiou et al., 2009, Wichterle et al., 2002) (Figure 1 —figure supplement 1).”

However, we believe this caveat is not of a major concern for the following reasons:

(a) The end goal of using these two different methods for labeling brachial MNs was to identify genes (transcription factors, terminal differentiation markers) that are expressed continuously in these cells. Our extensive validation of RNA-Seq results with RNA ISH suggests we met this goal (Figure 2-4, Figure 1 —figure supplement 2, new Figure 3 —figure supplement 1).

(b) As we mention in our reply to Reviewer 1 (please see R1 – Response 2), we conducted additional experiments to test the specificity of our genetic labeling approach (new Figure 1 —figure supplement 1).

(c) As we mention in Results, the RNA-Seq of genetically labeled brachial MNs was: (a) sensitive because it identified genes expressed in subtypes of brachial MNs (Ebf2, Lhx3, Irx), and (b) region-specific because it identified specific Hox genes, known to be expressed only in brachial MNs

Reviewer #1 (Recommendations for the authors):1. My major concern with this paper is the lack of specificity for the findings and ease in building a cohesive story. The paper is largely centered around RNAseq and RNA in situ (that is sometimes hard to interpret, see comment below). I appreciate the large amount of effort put into this study, but it would be nice to know how many of these gene regulatory networks are direct (ie. TF binding to consensus motifs in promoter regions). I was unable to follow a clear logic as to why certain genes were chosen for follow up, or to focus large parts of the story on.

We regret that in some parts the original version of the manuscript lacked cohesion. To address this, we have now made significant changes in Results and also removed all data on Irx transcription factors, as suggested by Reviewers 1 and 2. We hope the revised manuscript is now more focused and communicates effectively its main conclusion, i.e., sustained Hoxc8 activity is necessary for brachial motor neuron terminal differentiation.

To address the comment on lack of specificity, we have now performed RNA fluorescent in situ hybridization (FISH) coupled with a motor neuron-specific marker (FoxP1). These new findings (presented in Figure 3G and Figure 3 —figure supplement 1) corroborate the initial results reported in this paper.

Motivated by the reviewer’s comment to identify direct Hoxc8 target genes, we analyzed recently published RNA-Seq and ChIP-seq datasets from motor neurons derived from ES cells (ESC-MNs) in which Hoxc8 expression can be induced upon doxycycline treatment (Bulajic et al., 2020). Our analysis showed that the same Hoxc8 target genes (e.g., *Pappa, Glra2*) we found downregulated in brachial motor neurons of *Hoxc8 MNΔ ^early^* and *Hoxc8 MNΔ ^late^* mice are upregulated in motor neurons derived from ES cells (ESC-MNs), in which Hoxc8 expression is induced (Bulajic et al., 2020) (Figure 5A). Lastly, we found that Hoxc8 binds directly at the *cis*-regulatory region of the terminal differentiation genes (e.g., *Mcam, Pappa, Glra2*) we identified in vivo, and is sufficient to induce their expression in the context of ESC-MNs (Figure 5B, Figure 5 —figure supplement 1).

2. In regards to Figure 1, how specific are HB9::GFP and Chat::Ai9 to motor neurons, or neurons in general. Authors should co-stain for other neuronal and glial markers.

To our knowledge, the *HB9::GFP* mouse line is the best available tool to genetically label mouse embryonic motor neurons. This mouse line has been used by several recent studies for the same purpose and at the same time point with us: to isolate mouse spinal MNs at e12.5 with FACS and then conduct RNA-Sequencing (Sawai et al., *eLife,* 2022, PMID: 34994686; Hanley et al., *Neuron*, 2016, PMID: 27568519; Amin et al., *Science*, 2015, PMID: 26680198). However, we cannot exclude the possibility that a very small population of the HB9::GFP+ cells we isolated at e12.5 are spinal interneurons. The original paper (Wichterle et al., Cell, 2002, PMID: 12176325) describing the generation of *HB9::GFP* mice notes: “GFP expression was found in motor neuron cell bodies, dendrites, and axons in HB9::GFP embryos. In this line a very low level of expression (10- to 20-fold lower than in motor neurons) was detected in DRG neurons and a subset of ventral interneurons at e10.5, but not later”.

It is also unlikely that we isolated any glia at e12.5 with the *HB9::GFP* line – mature oligodendrocytes are not present at e12.5 in the spinal cord, as the peak of myelination in mice occurs at postnatal day 21 (p21). Moreover, the aforementioned studies (PMID: 34994686, PMID: 27568519, PMID: 26680198) have not reported any *HB9::GFP* expression in glia.

Prompted by the reviewer’s suggestion, we conducted additional experiments to evaluate the specificity of labeling spinal motor neurons at p8 with the *ChAT::IRES::Cre*; Ai9 [Rosa26-CAG^promoter^-loxP-STOP-loxP-tdTomato] mouse line. Double immunofluorescence staining using antibodies against ChAT (cholinergic motor neuron marker) and tdTomato (expressed in cells in which Cre is/was active) revealed robust co-localization of ChAT with tdTomato in most (if not all) spinal motor neurons (Figure 1 —figure supplement 1).

We also observed sparse tdTomato expression, but not ChAT, in a few cells (5-10 cells per section) located more dorsally and medially in the spinal cord, which are not motor neurons (Figure 1 —figure supplement 1). A recent study that used the same *ChAT::IRES::Cre* line reported similar observations and suggested that this is likely due to earlier Cre expression in the lineage of these tdTomato positive cells (PMID: 33931636).

At p8, there are many astrocytes present in the spinal cord but few mature oligodendrocytes (myelination peaks at p21). We therefore stained against the astrocyte marker GFAP (Millipore, AB5541, 1:200) and found no colocalization with tdTomato-expressing cells in *ChAT::IRES::Cre; Ai9* spinal cords at p8 (Figure 1 —figure supplement 1). We also attempted to stain for a microglia marker CD11b (Bio-Rad, MCA711, 1:50), but we did not detect any staining likely due to technical reasons. Altogether, these new data suggest that the tdTomato-expressing cells we isolated at p8 by FACS are mostly spinal motor neurons. Please, see also ER- Response 4 (page 3 of this document).

Also, why not use a reporter from one of the identified terminal differentiation factors which is on early and late, so that a single reporter is used?

This is a great suggestion but it would require the generation and characterization of new mouse lines, as well as redoing the RNA-Seq experiments at e12 and p8. Such endeavor will take 4-5 years to complete judging from the amount of time it took to complete the current study. However, we are planning on generating in the future new reporter/Cre lines based on the terminal differentiation markers identified from this study.

3. The RNA in situs in Figures3A, 4E, 5B and some of the supplemental images are not of the highest quality and one has to read the manuscript to know what to conclude. Perhaps newer approaches like RNAscope, or similar probes, would be more quantitative and easier to interpret. Also some images have white dotted circles around them cells of interest and other images do not – please be consistent.

We conducted new RNA fluorescent ISH (FISH) experiments coupled with a MN-specific marker (Foxp1). These new data corroborate our original conclusion: Hoxc8 controls the expression of several target genes in brachial MNs (new Figure 3G, new Figure 3 —figure supplement 1). To complement these in vivo findings, we now provide evidence in in vitro generated MNs that Hoxc8 is sufficient to induce the expression of the same target genes (e.g., *Pappa, Glra2, Sema5a*) we identified in vivo (new Figure 5A). Lastly, we are now consistent with the white dotted circles. We only use them when we show Hoxc8 antibody staining or Hoxc8 RNA ISH data because Hoxc8 is also expressed in other cells of the spinal cord. We decided against using white dotted circles in all figures in order to be minimally invasive and avoid excessive drawing on top of our microscopy images. There is, however, a spinal cord schematic next to each image to orient the reader on what part of the spinal cord is actually shown.

4. Although sympathetic to the need to use the Hoxc8 late mice for behavioral experiments due to perinatal lethality with the Hoxc8 early mice, these experiments demonstrate there is no phenotype and the author's assertion that mutant mice "tend" to perform worse is an unfair stretch. Moreover, with an N number of 3 and 5 mice on some of the experiments, its hard to make conclusions. The authors should consider consulting with or collaborating with a more behavior focused lab to do these experiments in a more robust manner, adding more tests and mice, or consider leaving the behavior experiments out of this paper.

Thank you for this excellent suggestion. By expanding our mouse colony, we have now conducted behavioral experiments both on *Hoxc8 MNΔ^early^* and *Hoxc8 MNΔ^late^* mice and increased N (new Figure 6, new Figure 6—figure supplement 1). Prompted by Reviewers 1 and 3, we have added a new test (treadmill analysis) on both *Hoxc8 MNΔ^early^* and *Hoxc8 MNΔ^late^* mice. Please see our detailed answer in ER – Response 1 (page 1 of this document).

Reviewer #2 (Recommendations for the authors):We have the following comments and suggestions for the authors to consider:1) Hox factors are known to control motor neuron subtype/pool specification (e.g. Dasen et al., 2005). It would be helpful to know if Hoxc8 early or late deletions change motor neuron subtype/pool identity rather than their maturation, which could be the reason for the differential gene expression This could be confirmed with co-labeling of reporter and subtype/pool markers, as in the Irx experiments.

This is an excellent point that we now clarified by text changes and new experiments.

A previous study (Catela et al., Cell Reports, 2016, PMID: 26904955) found that the overall organization of brachial MNs into columns is normal in *Hoxc8 MNΔ^early^* mice. However, the same study found that the expression of two motor neuron pool markers (Scip, Pea3) at e12.5 is partially affected, suggesting defects in motor neuron pool specification when *Hoxc8* is deleted early. In the current manuscript, we report that brachial motor neurons are generated in normal numbers in *Hoxc8 MNΔ^early^* mice (Figure 3C), but the expression of terminal differentiation genes (e.g., *Nrg1, Mcam, Pappa*) is affected in these mice (Figure 3F-G). Moreover, Hoxc8 binds directly on the *cis*-regulatory region of these genes (new Figure 5B, new Figure 5—figure supplement 1).

For the *Hoxc8 MNΔ^late^* mice, we attempted to stain brachial motor neurons with the motor neuron pool markers (Scip, Pea3) at e14.5 because at 14.5 we detect robust Hoxc8 depletion in these mice (new Figure 4B). However, these markers are transiently expressed and become downregulated in wildtype samples after e12.5. This prevented us from evaluating their expression at e14.5 in *Hoxc8 MNΔ^late^* spinal cords. Taken together, our analyses show that the expression of several terminal differentiation markers (e.g., *Nrg1, Mcam, Pappa, Glra2*) is affected in brachial MNs of *Hoxc8 MNΔ^late^* mice, which could explain their milder behavioral phenotype when compared to the *Hoxc8 MNΔ^early^* mice (new Figure 6, Figure 6 – videos 1 – 4).

In Results, we mention: “Together, these data show that Hoxc8 MNΔ^late^ mice display a milder behavioral phenotype compared to Hoxc8 MNΔ^early^ mice. This is likely due to the fact that Hoxc8 MNΔ^early^ mice display a composite phenotype (i.e., defects in early MN specification and axon guidance (Catela et al., 2016) combined with terminal differentiation defects [this study], whereas the Hoxc8 MNΔ^late^ mice only display terminal differentiation defects (this study). Although we cannot exclude the possibility that the terminal differentiation defects of MNΔ^early^ mice are a consequence of their early MN specification defects, this is unlikely as Hoxc8 binds directly to the cis-regulatory region of terminal differentiation genes (Mcam, Pappa, Glra2) (Figure 5B).”

2) Based on Figure 4F, the authors seem to distinguish terminal differentiation genes (e.g. neurotransmitters, synapse molecules) from axon guidance molecules and transcription factors. However, there are times when these seem to be treated interchangeably (e.g. discussion of Sema5a and Ret on page 12). The authors should more clearly and consistently state what they consider terminal differentiation markers in the text.

Thanks, we recognize the confusion and have modified the text accordingly in several occasions:

“Two simple, but not mutually exclusive mechanisms can be envisioned for the continuous expression of terminal differentiation genes (e.g., Slc10a4, Nrg1, Nyap2, Sncg, Ngfr, Glra2) in brachial MNs.”

“Lastly, Hoxc8 can only activate expression of Sema5a (member of Semaphorin family) at embryonic stages (Figure 3F-G, Table 1).”

“In this study, we uncovered several Hoxc8 target genes encoding different classes of proteins (Sema5a – axon guidance molecule; Glra2, Nrg1, Mcam, Pappa – terminal differentiation genes) (Figure 3E, Figure 4F), suggesting Hoxc8 controls different aspects of brachial MN development through the regulation of these genes.”

3) It would be worthwhile to have co-labeling with reporter (Hb9-GFP vs ChAT-tdTom) for at least some of the selected genes (e.g. in Figure 3G) to see if there is specific reduction in the cells of interest rather than just through correlation based on spatial pattern.

To address this important point, we coupled RNA fluorescence in situ (FISH) with antibody staining for FoxP1 (marker of LMC column in brachial region) on spinal cords of *Hoxc8 MNΔ^early^* mice. Consistent with what we originally reported, these experiments showed that the expression of several Hoxc8 target genes (*Sema5a, Mcam, Pappa*) is affected in brachial motor neurons (new Figure 3G; new Figure 3 —figure supplement 1). Moreover, most of the target genes (e.g., *Pappa, Sema5a, Glra2*) that are downregulated – in vivo – in motor neurons of *Hoxc8 MNΔ^early^* and *Hoxc8 MNΔ*
^late^ mice are upregulated upon inducible Hoxc8 expression in vitro generated motor neurons (new data in Figure 5A). Hence, Hoxc8 appears to be both necessary and sufficient for the expression of its target genes.

4) The Irx2 knockout experiments suggest that this downstream target of Hoxc8 may be involved in specification of limb-generating motor neurons, but not in the expression of terminal differentiation markers regulated by Hoxc8. This result seems tangential to the paper as 1) the knockouts are performed very early (prior to cell specification) and therefore does not provide any additional information about how Hoxc8 regulates motor neuron development past specification stage and 2) do not a clear functional link between Hoxc8 and terminal differentiation regulation.

We completely agree and have decided to remove the mouse Irx data from this manuscript. The manuscript is now more focused on Hoxc8 and its conclusions are strengthened by the addition of new molecular and behavioral data, two new main figures [Figure 5 – 6] and 5 new supplementary figures [Figure 1—figure supplement 1; Figure 3—figure supplement 1; Figure 4—figure supplement 1; Figure 5 —figure supplement 1; Figure 6 —figure supplement 1).

5) It is unclear what the *C. elegans* Irx2 (fosmid) experiments add to this paper. The loss of MNs entirely seems to detract from the overall point of this paper, which is that Hoxc8 helps to establish/maintain brachial motor neuron identity/maturation.

Again, we agree and have removed the *C. elegans* Irx data from the manuscript.

6) It is unclear what the behavioral experiments add to the paper since would be very difficult to attribute any phenotype to a specific cause. Also, unclear why the experiments were performed only in late Hoxc8 deletion and not the early deletion as well.

Please see ER – Response 1 (page 1 of this document). We conducted new behavioral tests (rotarod, forelimb grip strength, treadmill) on *Hoxc8 MNΔ^early^* and *Hoxc8 MNΔ*
^late^ mice. Our new behavioral data (new Figure 6) nicely complement the molecular analysis of brachial motor neurons in *Hoxc8 MNΔ^early^* and *Hoxc8 MNΔ*
^late^ mice.

7) The quality of some of the in situs should be improved (e.g. Glra2 in Figure 4E)

We have conducted new RNA FISH experiments to improve the quality of our findings (new Figure 3G; new Figure 3—figure supplement 1). Moreover, we performed double immunofluorescence (instead of RNA ISH) to demonstrate Hoxc8 protein depletion in brachial MNs of *Hoxc8 MNΔ*
^late^ mice (new Figure 4B).

8) It might be of interest to overexpress Hoxc8 in other regions of the spinal cord (e.g. cervical or thoracic) using chick electroporation for example to see if it can lead to overexpression of some of these terminal differentiation genes. This gain-of-function experiment may lend additional support to the proposal that Hoxc8 is required to maintain expression of these genes, although I am not sure that this is a requirement for distinguishing terminal selectors.

This is an excellent idea. In the absence of an established chick electroporation system in our lab, we analyzed recently published RNA-Seq and ChIP-seq datasets from motor neurons derived from mouse ES cells (ESC-MNs), in which Hoxc8 expression is induced with doxycycline (Dox) treatment (Bulajic et al., 2020, PMID: 33028607). We included this analysis in Results as shown below:

“Hoxc8 is sufficient to induce its target genes and acts directly

To gain mechanistic insights, we analyzed recently published RNA-Seq and chromatin immunoprecipitation-sequencing (ChIP-seq) datasets on MNs derived from mouse embryonic stem cells (ESC), in which Hoxc8 expression was induced with doxycycline (Bulajic et al., 2020). Our RNA-Seq analysis showed that induction of Hoxc8 (iHox8) resulted in upregulation of previously known (Ret, Scip/Pouef1) and new (Pappa, Glra2, Sema5a) Hoxc8 target genes (Figure 5A). Moreover, ChIP-seq for Hoxc8 in the context of these iHoxc8 ESC-derived MNs revealed binding in the cis-regulatory region of all these genes (Figure 5B), suggesting Hoxc8 acts directly to activate their expression. This in vitro data together with the in vivo findings in Hoxc8 MNΔ ^early^ and Hoxc8 MNΔ ^late^ mice (Figure 3F-G, Figure 3 —figure supplement 1, Figure 4C) suggest that Hoxc8 is necessary and sufficient for the expression of several of its target genes in spinal MNs.

Importantly, not all Hoxc8 target genes (e.g., Nrg1, Mcam) we identified in vivo are upregulated in iHoxc8 ESC-derived MNs (Figure 5 —figure supplement 1). This is likely due to the lack of Hoxc8 collaborating factors in these in vitro generated MNs. A putative collaborator is Hoxc6 because (a) Hoxc6 and Hoxc8 are co-expressed in embryonic brachial MNs (Catela et al., 2016), (b) animals lacking either Hoxc6 or Hoxc8 in brachial MNs display similar axon guidance defects (Catela et al., 2016), and (c) Hoxc6 and Hoxc8 control the expression of the same axon guidance molecule (Ret) in brachial MNs (Catela et al., 2016). Supporting the notion of collaboration, our analysis of available ChIP-seq data for Hoxc6 and Hoxc8 from iHoxc6 and iHoxc8 ESC-derived MNs(Bulajic et al., 2020), respectively, showed that these Hox proteins bind directly on the cis-regulatory region of previously known (Ret, Gfra3) and new (Mcam, Pappa, Nrg1, Sema5a) Hoxc8 target genes (Figure 5 —figure supplement 1).”

Altogether, our in vivo data and in vitro analysis suggest that Hoxc8 is necessary and sufficient for the expression of at least some its target genes. Indeed, most terminal selector-type TFs in *C. elegans* are both necessary and sufficient. Our findings on mouse Hoxc8 support the idea that key features of terminal selector function are conserved across species (please see last paragraph in Discussion).

Reviewer #3 (Recommendations for the authors):1. The authors used two different genetic systems to label brachial MNs at embryonic day 12 (e12) and postnatal day 8 (p8) as it was not possible for them to label MNs with just a single genetic system. Using these reagents they conclude that Hoxc8 regulates some of the same and some different targets. But can they rule out that some of these results is a consequence of using two different labeling systems? Are they certain that the cells labeled at both time points are the same cells? Perhaps using a lineage tracing tool and/or normalizing with some of the genes they discover in their RNA-seq experiments may be a way add confidence in the similarities and differences that they find in the RNAseq datasets are accurate.

Indeed, this is a very important point which we now address with text changes and new experiments. Please see our detailed responses in ER – Response 4 (page 3 of this document) and R1 – Response 2 (page 6). In brief, Hb9-GFP expression is faint at post-natal stages, which necessitated the use of an additional reporter strategy (ChAT-Cre; Ai9 tdTomato line) for the discovery of genes expressed in brachial MNs via RNA-Sequencing. In Results, we acknowledge the limitations of using two different genetic labeling strategies. We also conducted additional experiments to evaluate the specificity of motor neuron labeling by staining for glia and cholinergic motor neuron markers (new Figure 1 —figure supplement 1).

We note that the two different labeling systems were used in wild-type animals to identify enriched transcripts in embryonic and postnatal MNs (Figure 1). Hence, they do not affect our key conclusions on the function of Hoxc8 in motor neurons for the following reasons:

(a) The new Hoxc8 targets were discovered through RNA-Seq by using a single reporter (Hb9::GFP), not two different labeling systems, in *Hoxc8 MNΔ^early^* mice (Figure 3D).

(b) We used additional methods (i.e., RNA ISH) to independently validate our RNA-Seq results and identify new Hoxc8 target genes (Figure 3F, Figure 4C).

(c) We also provide new data to show that Hoxc8 controls terminal identity gene expression specifically in brachial MNs (with FoxP1) (Figure 3G, new Figure 3 —figure supplement 1).

(d) Our new analysis revealed that the same Hoxc8 target genes (e.g., *Pappa, Glra2, Sema5a*) we found downregulated in brachial motor neurons of *Hoxc8 MNΔ ^early^* and *Hoxc8 MNΔ ^late^* mice are upregulated in motor neurons derived from ES cells (ESC-MNs), in which Hoxc8 expression is induced upon doxycycline (Dox) treatment (Bulajic et al., 2020, PMID: 33028607) (new Figure 5, new Figure 5 —figure supplement 1).

2. Along the same lines, the RNA ISH of Hoxc8 at e12 looks dense, suggesting that all MNs in LMC and MMC regions express Hoxc8. On the other hand, expression of Hoxc8 seems sparse at p8, suggesting few Hoxc8+ MNs at p8 compared to e12 stage. Performing dual RNA ISH with Hoxc8 and endogenous genes of interest would provide greater confidence that the same number of Hoxc8+ MNs are present at two different time points.

Yes, the reviewer is correct. Consistent with our findings (Figure 3B), a previous study showed that, at e12, Hoxc8 is expressed both in MMC and LMC motor neurons (Catela et al., 2016, PMID: 26904955). Like most Hox genes, *Hoxc8* is also expressed in other neurons (not motor neurons) of the spinal cord (Figure 3B, 4B), which necessitated the need of Cre lines (Olig2::Cre, ChAT-IRES-Cre) to specifically inactivate it in motor neurons at early and late stages of development.

In wild type animals, spinal motor neurons are normally generated in excess and a large fraction of them (~50%) undergoes physiological/developmental cell death after e12, resulting in fewer motor neurons at later developmental stages (PMID: 10928282). Hence, the number of brachial motor neurons expressing Hoxc8 is high at e12 and ~50% lower at p8. To address the reviewer’s point, we conducted double immunofluorescence and RNA ISH. We found that Hoxc8 continues to be expressed in spinal motor neurons at late embryonic (e18.5) and early postnatal (p8) stages (new Figure 4 —figure supplement 1).

3. The authors state that Hoxc8 MN late mutants perform worse in rotarod performance test and forelimb grip strength in 3-month old mice. However, the statistical analysis suggests there is no significant change. It was unclear why the authors chose such a late time point for these assays, when the mice potentially have time to compensate for a compromised motor system. Redoing this experiment at earlier time points, such as with 1 month old mice (there is precedence for this in the literature), might reveal significant differences.

We have now conducted new behavioral tests (rotarod, forelimb grip strength, treadmill) both on *Hoxc8 MNΔ^early^* and *Hoxc8 MNΔ*
^late^ mice and significantly increased the number of animals. Please see ER – Response 1 (page 1 of this document) for a detailed discussion of the behavioral data, which nicely complement the molecular analysis of brachial motor neurons in *Hoxc8 MNΔ^early^* and *Hoxc8 MNΔ*
^late^ mice.